# Redesigning error control in cross-linking mass spectrometry enables more robust and sensitive protein-protein interaction studies

Boris Bogdanow [ID][1,2], Max Ruwolt[1], Julia Ruta [ID][1], Lars Mühlberg [ID][1], Cong Wang[1], Wen-feng Zeng[3,4], Arne Elofsson [ID][5] & Fan Liu [ID][1,6][✉]

## Abstract

**Cross-linking mass spectrometry (XL-MS) allows characterizing protein-protein interactions (PPIs) in native biological systems by capturing cross-links between different proteins (inter-links). However, inter-link identification remains challenging, requiring dedicated data filtering schemes and thorough error control. Here, we benchmark existing data filtering schemes combined with error rate estimation strategies utilizing concatenated target-decoy protein sequence databases. These workflows show shortcomings either in sensitivity (many false negatives) or specificity (many false positives). To ameliorate the limited sensitivity without compromising specificity, we develop an alternative target-decoy search strategy using fused target-decoy databases. Furthermore, we devise a different data filtering scheme that takes the inter-link context of the XL-MS dataset into account. Combining both approaches maintains low error rates and minimizes false negatives, as we show by mathematical simulations, analysis of experimental ground-truth data, and application to various biological datasets. In human cells, inter-link identifications increase by 75% and we confirm their structural accuracy through proteome-wide comparisons to AlphaFold2-derived models. Taken together, target-decoy fusion and context-sensitive data filtering deepen and fine-tune XL-MS-based interactomics.**

**Keywords** Cross-linking Mass Spectrometry; False-Discovery Rate; Proteomics; Structure Modeling; Error Control
**Subject Categories** Computational Biology; Proteomics

## Introduction

Cross-linking mass spectrometry (XL-MS) can reveal protein-protein interactions (PPIs) and the structural information of their binding interfaces in native biological systems (Liu et al, 2017; O'Reilly and Rappsilber, 2018). In XL-MS, protein contacts are captured using a cross-linker, a small organic molecule composed of a spacer arm and two reactive groups that typically target specific amino acid side chains, and are thereafter identified by mass spectrometry. Such experiments can yield intra-links (cross-links between residues within the same protein sequence), inter-links (cross-links between residues in different protein sequences), and mono-links (peptides that are modified by a partially hydrolyzed cross-linker). Only inter-links give insight into the structural configuration of PPIs.

The quality of XL-MS datasets critically depends on a balance between maximizing sensitivity (minimizing false negatives) and specificity (minimizing false positives). The same challenge is prevalent in standard bottom-up proteomics and is primarily addressed by estimating false-discovery rates (FDR) using a concatenated target-decoy search strategy (Elias and Gygi, 2007). In this approach, mass spectra are searched against a database consisting of the target protein sequences and a concatenated list of reversed, shuffled, or randomized versions of them, called decoys (Elias and Gygi, 2007). Any spectral matches to the decoys are assumed to mimic false positives, which allows statistical control of error rates via the FDR. FDR filtering can be applied separately for different subsets of spectral matches when the error rates between the subgroups are reasonably different. Such or similar strategies are used, for example, for modified (Marx et al, 2013; Fu and Qian, 2014), miscleaved, or differently sized peptides in shotgun proteomics (Cox and Mann, 2008).

The XL-MS field has widely adopted the concatenated target-decoy strategy for FDR filtering (Yang et al, 2012; Liu et al, 2015; Maiolica et al, 2007; Walzthoeni et al, 2012; Lenz et al, 2021; Fischer and Rappsilber, 2017; Crowder et al, 2023; Zhou et al, 2023). Initial FDR filtering approaches merged inter-links and intra-links into one category (inter-intra combined FDR). Nowadays, separate FDR filtering for intra- and inter-links (inter-intra separate) has become the prevailing strategy because inter-links have a higher error probability (Walzthoeni et al, 2012; Lenz et al, 2021). This subgroup-specific FDR filtering enables more accurate

[1]Research group "Structural Interactomics", Leibniz Forschungsinstitut für Molekulare Pharmakologie, Robert-Rössle-Str. 10, 13125 Berlin, Germany. [2]Institute of Virology, Campus Charité Mitte, Charité-Universitätsmedizin Berlin, corporate member of Freie Universität Berlin and Humboldt-Universität zu Berlin, Berlin, Germany. [3]Department of Proteomics and Signal Transduction, Max Planck Institute of Biochemistry, Martinsried, Germany. [4]Center of Infectious Disease Research, School of Engineering, Westlake University, 310024 Hangzhou, China. [5]Stockholm Bioinformatics Center, Stockholm University, SE-106 91 Stockholm, Sweden. [6]Charité—Universitätsmedizin Berlin, Charitépl. 1, 10117 Berlin, Germany. [✉]E-mail: fliu@fmp-berlin.de

FDR estimation for inter-links and PPIs. However, to what extent this approach impacts sensitivity in identifying inter-links has not been addressed.

Additional subgrouping strategies for FDR filtering proposed in recent studies increase the sensitivity toward inter-links by incorporating different levels of information present in the XL-MS dataset (e.g., about the specific cross-link, the individual protein, and the PPI). For instance, subgrouping of inter-links based on whether the connected proteins are additionally supported by intra-links or mono-links was shown to reduce the frequency of decoys when searching a concatenated target-decoy database (Chen et al, 2022; Sailer et al, 2022). Related concepts are employed by recently developed XL-MS search engines that enable more sensitive inter-link identification owing to aggregate scoring functions that integrate protein, PPI and cross-link level information (Crowder et al, 2023; Zhou et al, 2023). While such contextual information appears to be highly useful, it has not been systematically assessed whether the commonly used FDR control strategies in XL-MS (i.e., target-decoy concatenation) are sufficiently robust when performing context-sensitive strategies.

Here, we find that concatenated target-decoy searches can drastically underestimate the FDR when applying context-sensitive filtering. Instead, we demonstrate that the theoretical error is better approximated by a target-decoy fusion strategy. We leverage this observation to devise alternative inter-link subgrouping strategies that boost inter-link identifications by 29–76% in deep XL-MS datasets while simultaneously maintaining low error rates and increasing coverage of AlphaFold2 (AF2)-Multimer models across the proteome.

## Results

### FDR filtering with traditional data grouping schemes limits either the specificity or sensitivity of inter-link identification

In order to evaluate the sensitivity and specificity of different data grouping schemes used for FDR filtering, we made use of our recently developed XL-MS benchmarking standard. This dataset consists of 256 purified proteins that have been mixed and cross-linked according to a defined scheme (Clasen et al, 2024) prescribing which inter-protein connections will occur and which ones cannot occur (Fig. 1A). Following database search, we can accordingly assign cross-link (that is, unique residue pair) identifications as true or false, and assess the proportion of false-positive and false-negative identifications at any given target-decoy based FDR cutoff (Fig. 1B,C).

We initially tested the two most established strategies in XL-MS: a concatenated target-decoy FDR filtering strategy considering intra-links and inter-links either as one combined (intra-inter combined) or two separate (intra-inter separate) groups (Fig. 1D). In accordance with previous reports (Walzthoeni et al, 2012; Lenz et al, 2021), we find that the combined strategy fails to appropriately control error among inter-links, leading to many false-positive identifications (Fig. 1E). While this problem can be adequately controlled by the intra-inter separate approach, this strategy results in a considerable loss of sensitivity signified by an increase in false-negative identifications (Fig. 1F). For example,

approximately half of the available true matches remain unidentified at 1% FDR. These shortcomings of the traditional data grouping schemes in XL-MS highlight the need for a workflow that improves sensitivity toward inter-links while retaining accurate error estimates.

### Context-sensitive inter-link subgrouping schemes are not fully compatible with concatenated target-decoy searches

Previous work in shotgun proteomics found that peptide identifications can be increased by subgrouping strategies (Marx et al, 2013; Cox and Mann, 2008; Fu and Qian, 2014). This concept is based on the assumption that different subsets of matches have different error probabilities. Adapted to XL-MS, subgroups may be defined based on various properties of the detected inter-linked peptides, including peptide length, charge state, missed cleavages or the presence of other cross-links supporting the inter-link. We tested the latter possibility by separating inter-links depending on the extent to which other cross-links support the existence of the protein or PPI (context-rich vs context-poor inter-links). We evaluated two definitions of context-richness for their suitability as subgrouping criteria: the first depends on the presence of contextual intra-links ("intra-dependent") and has been proposed previously in a similar manner (Chen et al, 2022; Sailer et al, 2022), the second depends on the presence of contextual inter-links ("inter-dependent") and has been devised in this study (Fig. 2A). In both subgrouping procedures, inter-links are separated in the two subgroups prior to FDR analysis, which is then performed separately on both subsets before aggregating them into a combined list (see "Methods").

We tested intra-dependent and inter-dependent subgrouping combined with concatenated target-decoy searches on our XL-MS benchmarking data (Fig. 2B,C). Compared to the established intra-inter separate strategy, our inter-dependent subgrouping yields approximately the same fraction of false positives while substantially lowering the fraction of false negatives. Thus, the inter-dependent strategy appears to be an effective way to increase identification sensitivity without compromising specificity. An increase in sensitivity (i.e., reduction of false negatives) is also observed upon intra-dependent subgrouping (Fig. 2C). However, intra-dependent subgrouping leads to a massive increase in false positives, suggesting that error rate cannot be effectively controlled when using this subgrouping approach together with concatenated target-decoy databases.

To assess whether these observations hold true in a real-world scenario, we analyzed a deep XL-MS dataset from HEK293T cells that was recently generated in our lab (Clasen et al, 2024). This dataset was created by cross-linking with the enrichable cross-linker DSBSO (Wheat et al, 2021; Matzinger et al, 2020) ("HEK293T"), which yielded 12,216 target and 2003 decoy inter-links when no FDR cutoff was applied. When intra-dependent or inter-dependent subgrouping was applied, we observed divergent score distributions between context-rich and context-poor matches (Fig. EV1A,B), which was more pronounced when inter-dependent grouping was applied. This reinforces our initial assumption of different error probabilities between context-poor and context-rich subgroups and suggests that inter-link identification in the HEK293T dataset can benefit from context-sensitive subgrouping.

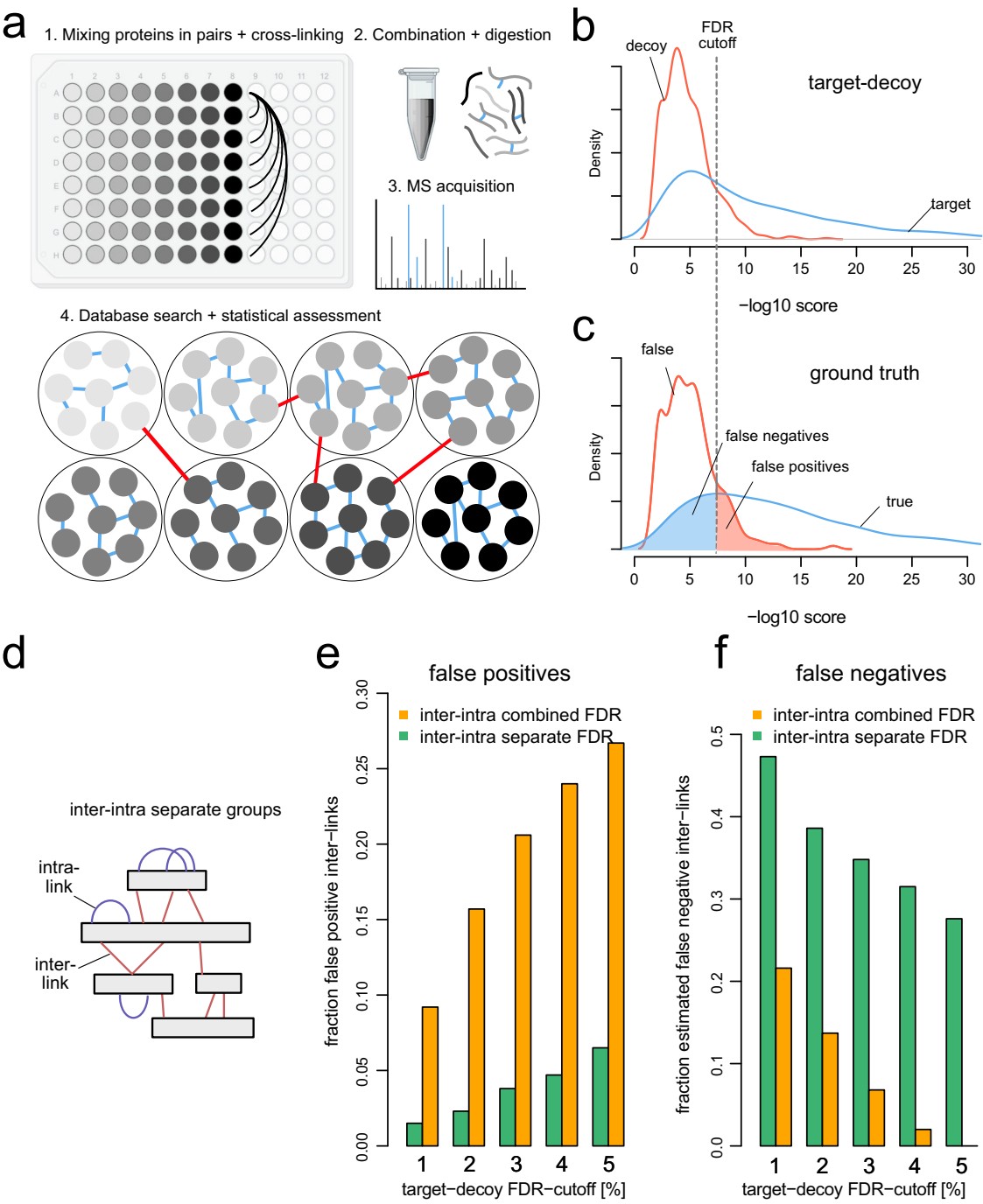

**Figure 1. Existing XL-MS data grouping strategies limit either the specificity or sensitivity of inter-link identification.**

(**A**) Workflow for data of ground-truth dataset generated by Clasen et al, 2024. Following pairwise mixing of recombinant proteins within one interaction group, proteins were denatured, cross-linked, followed by MS analysis. (**B**, **C**) The ground-truth dataset was searched using XlinkX. Database search score distribution of inter-link matches to target or reverted decoy entries (**B**) as well as true and false matches according to the ground-truth (**C**). FDR is estimated based on target-decoy competition, and an arbitrary FDR cutoff is indicated for illustration. Matches below and above the cutoff are classified as false negatives or false positives according to the ground truth. (**D**) Schematic of cross-link identification types in XL-MS. Intra-links match within the same protein sequence, while inter-links connect different protein sequences. (**E**, **F**) Fraction of false positives (**E**) or estimated false negatives (**F**) when considering intra-links and inter-links as one combined or two separate groups in concatenated target-decoy FDR filtering strategies.

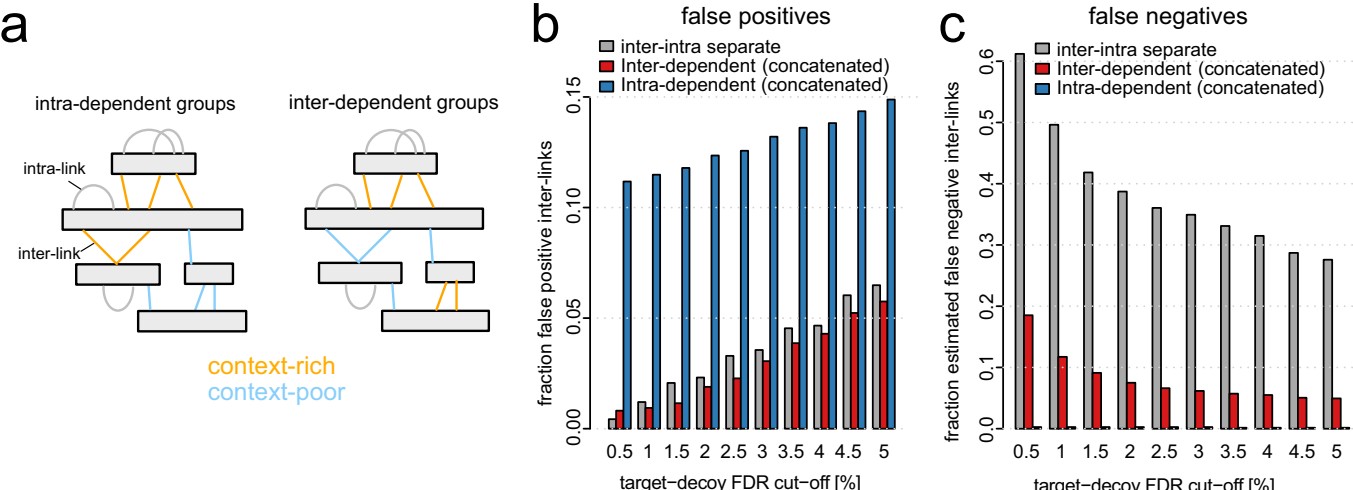

**Figure 2. Defining context-sensitive subgrouping strategies.**

(A) Design of context-sensitive subgrouping. Upon intra-dependent subgrouping (left), inter-links were considered as a separate, context-rich subgroup when both corresponding proteins were additionally matched by intra-links. All other inter-links are grouped as context-poor. Upon inter-dependent subgrouping (right), inter-links are grouped as context-rich when at least one additional inter-link connects the same protein pair via different lysines. All other inter-links are grouped as context-poor. (B, C) Fraction of false-positive (B) or estimated false-negative (C) inter-links when performing inter-intra separate or the two subgrouping strategies.

We first separated inter-links according to the grouping criteria and assessed the overall fraction of targets and decoys prior to imposing any FDR cutoff (Fig. 3A). We observed that both inter-link and intra-link subgrouping strategies result in very few decoys among context-rich subgroups. For inter-dependent subgrouping, this observation agrees with the results from our ground-truth dataset. However, for intra-dependent subgrouping, the complete lack of decoys among the context-rich inter-links is in stark contrast to the explosion of false positives we observed in the benchmarking dataset. Following the herein-applied FDR filtering strategy using target-decoy concatenated database, more than 10,000 inter-links could be added to the identification without accepting any false match. The inconsistency to our benchmarking dataset indicates that combining intra-dependent subgrouping and concatenated target-decoy searches might fail to recognize the false positives among the context-rich inter-links.

Since the true number of false positives in the HEK293T data cannot be known, we simulated the theoretically expected distribution of true and false positives in the context-rich and context-poor subgroups based on a set of simple assumptions and controlled parameters (see "Methods"). We assume that wrong matches are randomly assigned to any protein in the database, while true matches have the tendency to frequently match to a restricted set of proteins. Following the placement of correct and incorrect matches on the proteins, we grouped context-poor and context-rich inter-links and evaluated the fraction of correct and incorrect matches in these subgroups. This is advantageous, as a simulation allows direct counting of the number of correct and incorrect matches (Fig. EV2). For intra-dependent subgrouping, the simulation gives ~1300 false positives representing ~11.9% of the identifications in the context-rich subgroup (Fig. 3B), which is in stark contrast to the experimentally observed numbers of decoys in this subgroup (see Fig. 3A). Interestingly, the simulated distribution of false positives for the inter-dependent strategy closely resembles

the distribution of decoys. Altogether, this suggests that the error rate can be properly controlled for inter-dependent subgrouping, but not for intra-dependent subgrouping.

The problem with intra-dependent subgrouping is that a cross-link group with a low error rate (intra-links) is used to categorize a cross-link group with a substantially higher error rate (inter-links). This becomes intuitively apparent when considering how intra-dependent filtering is impacted by having high numbers of intra-links in settings of restricted database size: When more and more intra-links are identified, many (if not most) proteins in a database will eventually contain an intra-link. Randomly assigned false-positive inter-links will then more frequently contain proteins supported by these intra-links (Fig. 3C). This issue is avoided in inter-dependent subgrouping because this strategy categorizes inter-links based on information from the same cross-link group. Inter-dependent subgrouping builds on the fact that inter-links belonging to PPIs, which are additionally targeted by different inter-links are very unlikely to occur by chance.

## Target-decoy fusion enables accurate error control upon context-sensitive inter-link subgrouping

Based on the explosion of false-positive identifications in the benchmarking and simulated datasets, we reason that the robustness of target-decoy concatenation strategies can be compromised whenever higher-level information (e.g., proteins supported by intra-links) is used to classify lower-level matches (e.g., specific inter-links). Integration of different information levels can also occur in conventional bottom-up proteomics and has been addressed by the introduction of fused target-decoy databases (Savitski et al, 2015; Zhang et al, 2012). In this setup, each decoy entry is fused to its respective target entry, creating one fused protein sequence (Fig. 3D). In an ungrouped setting, the distribution of target and decoy hits is not different when using a

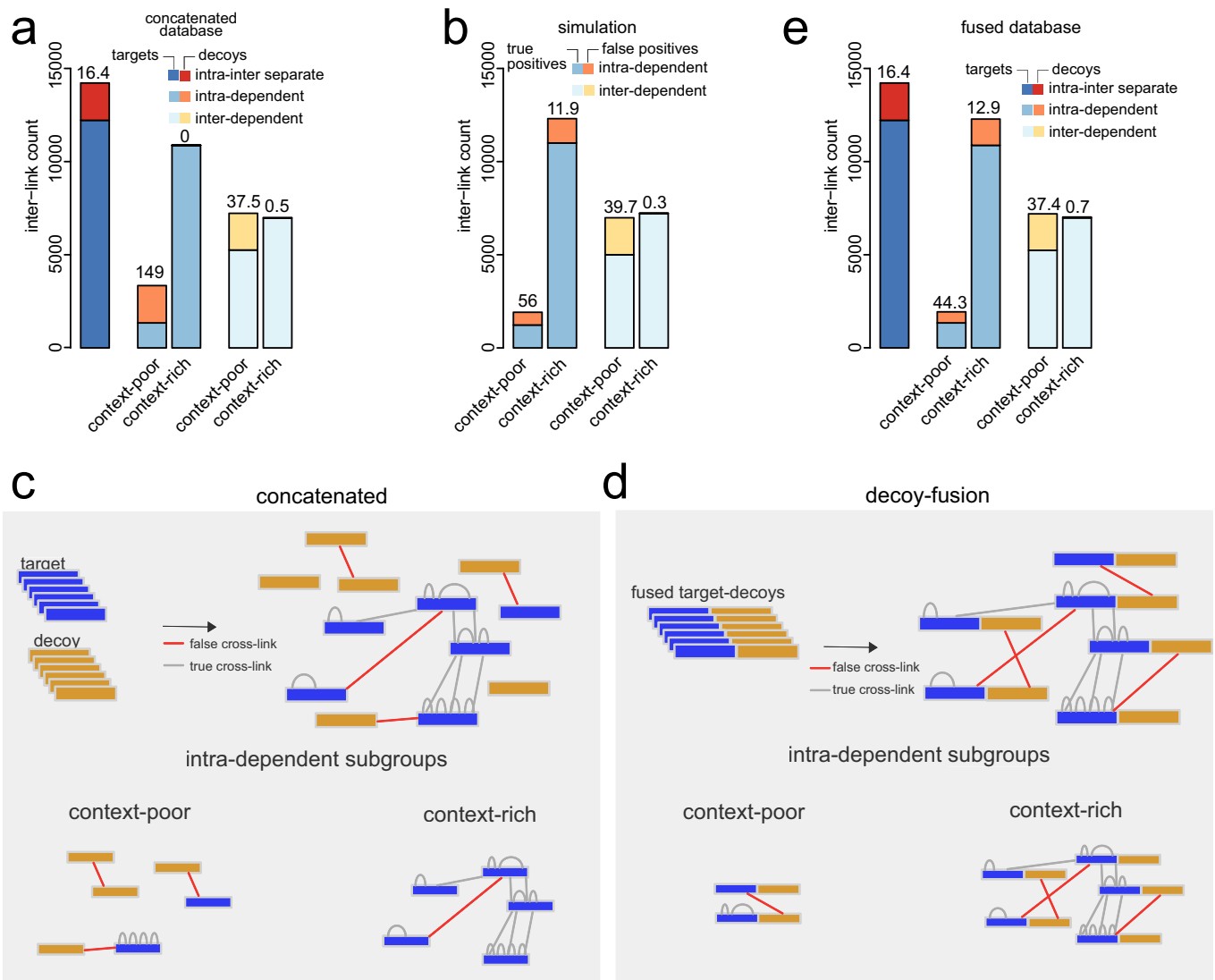

**Figure 3. Target-decoy fusion enables context-sensitive strategies in XL-MS.**

(A) Distribution of inter-link matches to target or decoy entries in deep proteomic XL-MS "HEK293T" dataset upon intra-inter separate grouping or further subgrouping of inter-link matches according to intra- and inter-dependent grouping criteria using a concatenated target-decoy database. No FDR cutoff was applied. (B) Simulated distribution of true and false-positive matches to context-rich and context-poor subgroups upon context-sensitive subgrouping of inter-links. (C, D) Schematic illustration of false and true assignments in concatenated (C) and fused (D) database search strategies. (E) Same analysis as in (A), using a fused target-decoy database. For panels (A, B, E), the number on top of the bars corresponds to the fraction of decoys to targets (A, E) or false to true identifications (B) in percent.

concatenated or fused decoy database. However, upon context-sensitive subgrouping, fused databases ensure that once a target protein is assigned to a context-rich subgroup its corresponding decoy part is automatically assigned to the same subgroup.

We hypothesized that a fused decoy strategy will also be beneficial for context-sensitive subgrouping in XL-MS. Repeating our HEK293T dataset analysis with this strategy resulted in a realistic decoy count for all subgrouping strategies including the intra-dependent approach (Fig. 3E). Similarly, reanalysis of our benchmarking data shows that target-decoy fusion can reduce the false-positive identifications for intra-dependent subgrouping to similar levels as those observed for inter-dependent subgrouping and the inter-intra-separate strategy (Fig. 4A). Comparing all

strategies, we find that inter-dependent subgrouping is best suited for minimizing false positives (Fig. 4A) and false negatives (Fig. 4B).

## Combining inter-dependent subgrouping and target-decoy fusion boosts interactome coverage in various biological systems

Next, we sought to explore the benefits of context-sensitive subgrouping and fused target-decoy searches when studying biologically relevant interactomes. To this end, we analyzed interactome coverage in intact human cells using the HEK293T dataset described above, intact human mitochondria ("mito") (Zhu

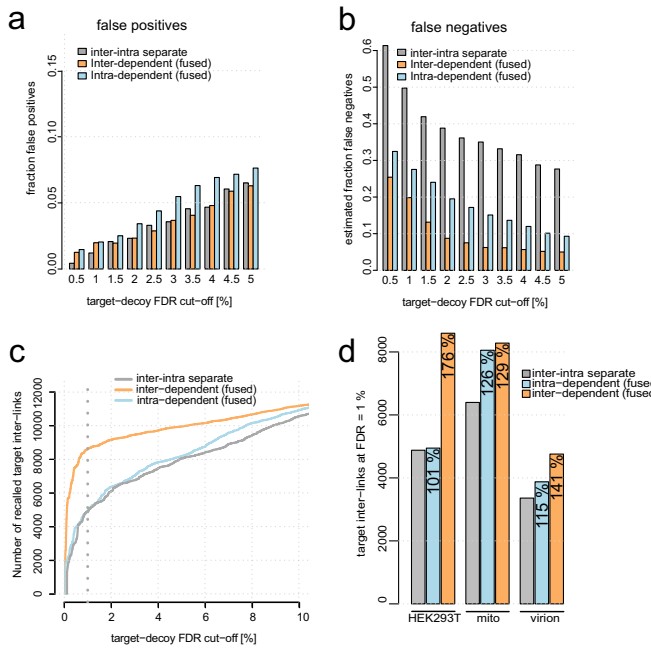

**Figure 4. Target-decoy fusion improves statistical power in inter-link assignments.**

(A, B) Fraction of false-positive (A) and estimated false-negative (B) inter-links in the ground-truth dataset (Clasen et al, 2024) for different data subgrouping strategies and FDR cutoffs. (C) Recall of target inter-links as a function of the FDR cutoff comparing different data grouping strategies for the HEK293T dataset. (D) Recall at 1% FDR for three different datasets and data grouping strategies with percent improvement compared to the inter-intra separate approach. All analyses were performed using fused target-decoy databases.

et al, 2024), and intact herpesviral particles ("virion") (Bogdanow et al, 2023). We first used the simulation approach described above to confirm the utility of target-decoy fusion databases and context-sensitive subgrouping also for the mito and virion datasets (Fig. EV3A–G). In all cases, only target-decoy fusion accurately reflects simulated error rates. Context-rich subgroups, particularly the inter-dependent subgroup, display low error rates in all systems, which also holds true at the level of cross-link spectrum matches (Fig. EV3H–J) The inter-dependent context-rich subgroup further contained overall relatively few unique proteins, suggesting that the majority of the cross-links within this subgroup match to a restricted set of confident proteins (Fig. EV4A–C). As discussed above, this can be explained by the low probability of two different random (i.e., false) inter-links coincidentally matching to the same two protein sequences, which becomes exceedingly rare as the search space expands.

To assess intra-dependent and inter-dependent subgrouping for their potential to increase sensitivity (and thereby interactome coverage), we counted the number of identified inter-links that could be assigned at any FDR. We compared these results to those obtained when grouping all inter-links together (intra-inter-separate). In all tested datasets, we observed a pronounced increase in statistical power when inter-dependent grouping is applied (Fig. 4C and EV5A,B). The improvement was smaller (for "mito" and "virion" datasets) or non-existent (for "HEK293T") upon intra-dependent grouping. In all cases, inter-dependent grouping

performed best, particularly for stringent FDR cutoffs (29–76% increase in sensitivity at FDR = 1%) (Fig. 4D). These data show that context-sensitive subgrouping on a fused decoy database is a universally applicable strategy to increase sensitivity while maintaining accurate FDR estimates, whereby inter-dependent subgrouping increases sensitivity more robustly and effectively than intra-dependent subgrouping.

## Increased coverage of PPI contact sites in proteome-wide AF2 models

Finally, we assessed the utility of our optimized inter-link identification strategy for structural biology. In the HEK293T dataset, compared to the standard inter-intra separate strategy, our inter-dependent filtering strategy increased inter-link counts for 38% of PPIs (Fig. 5A). We then mapped inter-links from both strategies onto proteome-wide predictions of complexes by AF2 multimer (Jumper et al, 2021). While both strategies resulted in a similar fraction of inter-links satisfying the computational model, we find that inter-dependent grouping increased the number of inter-links in models of good (pDockQ >0.5), acceptable (0.23 <pDockQ <0.5) and poor (pDockQ <0.23) quality (Fig. 5B). Focusing only on models of acceptable and good docking quality, we observed particularly robust increases of coverage in regions within the models that are likely disordered (pLDDT <50) (Fig. 5C,D), indicating that increased coverage is particularly prominent for regions of higher structural complexity. Overall, compared to the standard approach, inter-dependent subgrouping resulted in a similar distribution of Cα-Cα distances in good, moderately and poorly satisfied models (Fig. 5E). The newly identified inter-links provide additional supporting structural information for AF2-multimer models, such as the SMC2-SMC4 dimer (Fig. 5F, 10 additional cross-links) and large cryoEM structures, such as the proteasome (Fig. 5G,H, 17 additional cross-links). Thus, inter-dependent subgrouping increases the coverage of PPI contact sites, providing additional cross-linking data to support computational and experimental structural models across the proteome.

## Discussion

In this study, we benchmarked existing FDR filtering strategies on a ground-truth dataset. Our analysis confirms previous findings that inter-links should be separately analyzed from intra-links to limit the extent of false positives among the biologically more valuable inter-links (Fig. 1E). While effectively controlling error among inter-links, we find that this strategy leads to a critical reduction of sensitivity in inter-link identifications (Fig. 1F) resulting in low coverage of PPI sites. Importantly, we find that this loss can be overcome by subgrouping inter-links that have different likelihoods of being wrong (Fig. 4B).

We grouped inter-links into either a context-rich or a context-poor subgroup that are based on the existence of other matches supporting the existence of the protein or the PPI. To appropriately apply context-sensitive subgrouping, it is essential to assure that target-decoy symmetry is not violated. Initially, with the use of target-decoy concatenated databases, such violation happened upon intra-dependent subgrouping (Chen et al, 2022; Sailer et al, 2022) leading

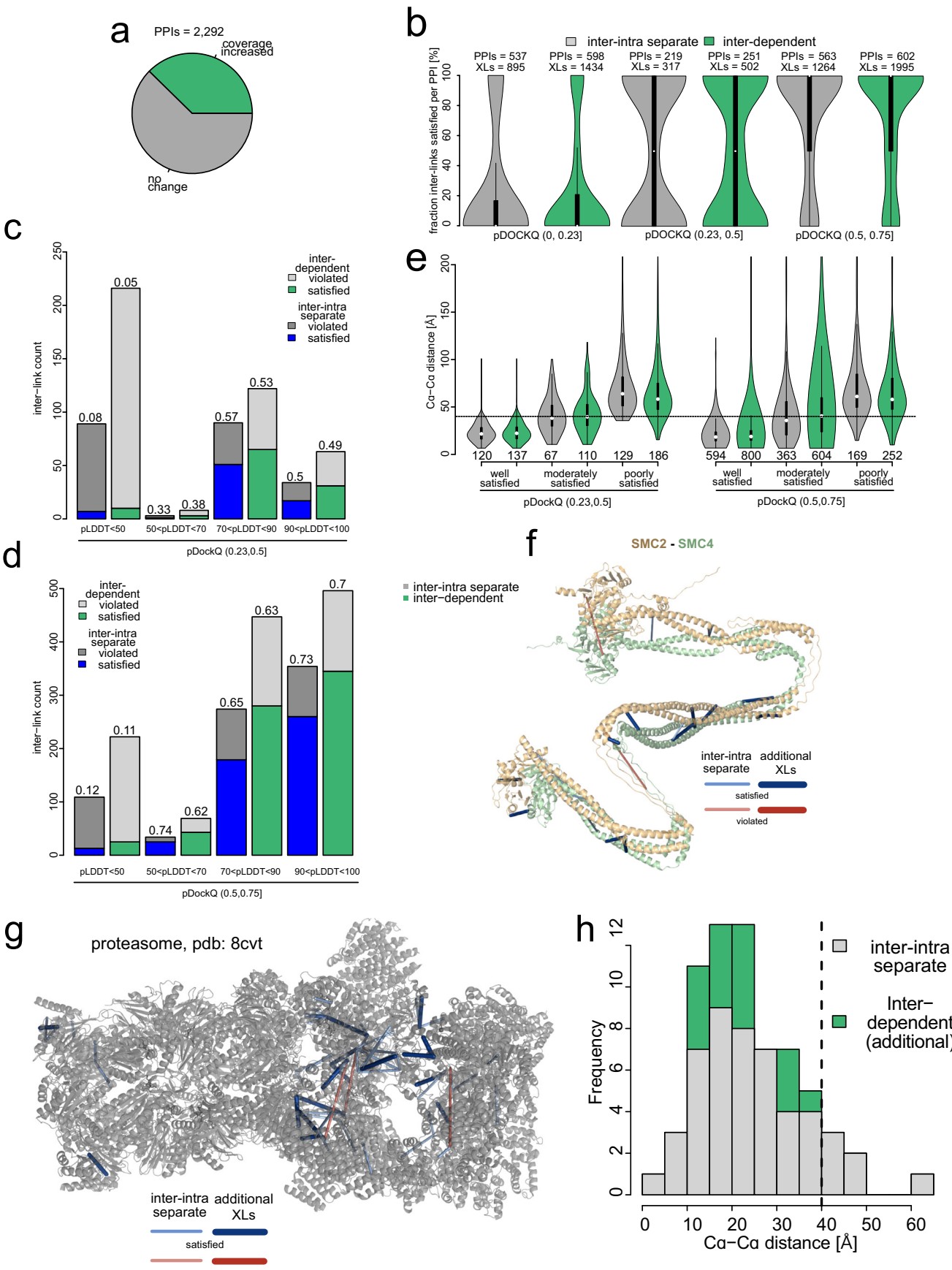

**Figure 5.   Inter-dependent subgrouping improves coverage of PPI contact sites.**

(A) Differences in inter-link coverage of PPIs between inter-dependent subgrouping and inter-intra separate grouping based on aggregating inter-links to PPIs. (B) Fraction of inter-links per PPI satisfying the distance constraint comparing both strategies for poor, moderate and well-predicted models. A pLDDT cutoff of 50 was applied for both cross-linked Lysines. The numbers above the violin plots corresponds to the number of protein-protein interactions (PPIs) or inter-links (XLs) (center point=median; box limits=upper and lower quartiles; whiskers = 1.5× interquartile range; outliers are minima and maxima of the distribution). (C, D) Number of Inter-links violating and satisfying the cross-linker distance constraint of 40 Å comparing inter-dependent subgrouping and standard approach (intra-inter separate) within AF2 models of (C) acceptable or (D) good quality. Different prediction quality at cross-linked residues (pLDDT, on both Lysines) was considered. (E) Models were binned according to their agreement with XL-MS data from the inter-intra separate search (well satisfied: 80–100%, moderately satisfied: 20–80%, poorly satisfied 0–20%). Inter-link distance comparing both strategies and model subsets is depicted. A pLDDT cutoff of 50 was applied for both cross-linked Lysines. The number below the violin plots corresponds to the number of inter-links included (center point=median; box limits=upper and lower quartiles; whiskers=1.5× interquartile range; outliers are minima and maxima of the distribution). (F) Example of AF2-predicted SMC2-SMC4 heterodimer with Inter-links identified in an inter-intra separate search and additional inter-links identified upon inter-dependent subgrouping. (G) Experimental cryoEM model of the proteasome and matching inter-links in an inter-intra separate search and additional links obtained from inter-dependent grouping. (H) Histogram of distances obtained from (G). An inter-link level FDR of 0.01 was set for both grouping approaches.

to the deceiving case of decoys being absent in the context-rich subgroup (Fig. 3A,B). In intra-dependent subgrouping, the context-rich subgroup consists of all inter-links where the corresponding proteins are additionally matched by intra-links. As intra-links have comparably low error probability (Lenz et al, 2021), they rarely match to decoys. In contrast, inter-links have much higher error probability and are more frequently present on decoys. Thus, while incorrect inter-links on target proteins coincide with correct intra-links to the same target proteins, incorrect inter-links on decoys rarely coincide with intra-links on the same decoy proteins. This leads to an underrepresentation of decoys in context-rich subgroups, violating target-decoy symmetry. To solve this issue, we adapted the decoy fusion database search design which was first introduced in shotgun proteomics workflows (Savitski et al, 2015; Zhang et al, 2012) and demonstrated its capabilities in restoring target-decoy symmetry and allowing better control of error rates (Fig. 3D,E). This is consistent with a similar strategy by the Rappsilber lab to restore error control of the mi-filter (Fischer and Rappsilber, 2024). The use of decoy fusion databases upon context-sensitive subgrouping is critical as it will model the error rate more faithfully. Decoy fusion ensures that both the target and decoy complements are placed in the same subgroup, whereas in concatenated strategies, they might end up in different subgroups.

While intra-dependent subgrouping on a fused database led only to modest improvements in inter-link detection sensitivity, we observed that our newly proposed subgrouping strategy, inter-dependent subgrouping, almost completely eliminated false negatives in the ground-truth dataset (Fig. 4B) and retained stringent error control (Fig. 4A). Furthermore, we demonstrated substantial improvements in PPI site coverage at stringent FDRs (Fig. 4C,D), allowing the generation of more comprehensive and highly reliable XL-MS-based interactomes. The reasons behind this excellent performance are the divergent error rates (Fig. 3E) and score distributions (Fig. EV1B) of inter-links between context-rich and context-poor subgroups. While the context-poor subgroup contains many "one hit wonders" where inter-links are the sole representative of the respective PPIs, context-rich inter-links consist of protein pairs that are found with different lysine connections. The latter type is unlikely to occur by chance as random cross-links are unlikely to coincidentally connect different lysines of the same protein pair.

XL-MS data contains more levels (e.g., cross-link spectrum match, residue pair, and PPI) and context-based (e.g., inter-link and intra-link) information compared to the data of shotgun

proteomics. Because of its inherent and complex structure, we expect that our fused target-decoy database design will augment the emergence of different context-sensitive strategies in XL-MS identification (Crowder et al, 2023; Chen et al, 2022; Zhou et al, 2023). Fused target-decoy strategies may also be used to consider other non-XL-based protein information, such as protein abundance to increase sensitivity. Future work may be directed towards exploring other subgrouping criteria or aggregating scoring functions that integrate multiple levels of information to continue pushing the boundaries of system-wide structural PPI profiling.

## Methods

### Reagents and tools table

| Reagent/resource | Reference or source | Identifier or catalog number |
|---|---|---|
| **Software** | | |
| R software, v.4.1.2. | https://cran.r-project.org/ | |
| AlphaFold multimer v.2.2 | Evans et al, 2022, https://doi.org/10.1101/2021.10.04.463034, https://github.com/google-deepmind/alphafold | |
| Rstudio, v. 2024.9.0.375 | https://github.com/rstudio/rstudio | |
| Pymol ™ Molecular Graphics System, Version 2.4.0a0. | https://www.pymol.org/ | |
| XlinkX v2.0 | Liu et al, 2017, https://doi.org/10.1038/ncomms15473 | |
| Proteome Discoverer v.2.5 | Thermo Fisher Scientific Inc. | |

### Reanalysis of datasets

We reanalyzed raw files (.raw) from Clasen et al (Clasen et al, 2024), containing HCD-MS2 spectra of the HEK293T and ground-truth datasets (ProteomeXchange identifier PXD042173). First, we converted into peak lists (.mgf files) in Proteome Discoverer v2.5 (Thermo Fisher). The.mgf files were used as input to identify cross-linked peptides with a stand-alone search engine based on XlinkX

v2.0 (Liu et al, 2017). The following settings of XlinkX were used: MS ion mass tolerance, 10 parts per million (ppm); MS2 ion mass tolerance, 20 ppm; fixed modification, Cys carbamidomethylation; variable modification, Met oxidation; enzymatic digestion, trypsin; and allowed the number of missed cleavages, 3; DSSO cross-linker, 158.0038 Da (short arm, 54.0106 Da; long arm, 85.9824 Da), reaction site: lysine, protein N-termini.

The search was performed using a database containing all 256 mixed, 264 additional entrapment, and 520 reversed decoy proteins. All cross-links matching to a database of 123 potential contaminant entries were removed prior to data analysis. True and false were assigned at the peptide level to account for the partial homology of proteins in different groups. True matches were assigned, when in accordance to the mixing scheme and false matches when in violation. It is important to note that, in principle, true matches may occur randomly when a cross-link matches two proteins within the same group. In this case, a true match is falsely assigned. However, the frequency of this is negligible as within group protein count ($n = 8$) is far lower than that of outgroup proteins ($n = 512$).

Raw files from HEK293T dataset (dowloaded from ProteomeXchange, identifier: PXD042173) containing HCD-MS2 data were converted into peak lists (.mgf) and searched as described above with DSBSO specificity (DSBSO cross-linker, 308.0388276 Da, short arm, 54.01056 Da, long arm, 236.01770 Da, reaction site: lysine, protein N-termini) with MS2 spectra searched against a concatenated target-decoy databases (randomized decoys) generated based on the corresponding proteome determined by bottom-up proteomics, containing 4860 target sequence entries. The table giving targets and decoy residue-level cross-links at no FDR threshold was used for subgrouping, FDR calculations and input for simulations. Cross-links refer to unique residue pairs, and the top scoring cross-link spectrum (CSM) match was considered as a representative for the residue pair.

For the mitochondrial (DSSO) dataset (Zhu et al, 2024) and the virion (DSSO) dataset (Bogdanow et al, 2023), the XlinkX search output available from the PRIDE deposition was used directly (under ProteomeXchange identifiers PXD032132, PXD031911).

## Target-decoy fusion strategy

In order to evaluate the performance of a fused decoy database search strategy, we analyzed our datasets as if they were searched using a fused database. To this end, we performed the database search in a concatenated design but then replaced the entries containing the gene names for the cross-linked proteins with their target annotations in the cross-link results table. For example, a cross-link matching the proteins "RPL12-RPS6(decoy)" was re-annotated as "RPL12-RPS6". Thus, all targets and decoys originating from the same .fasta entry are fused into one entry. Importantly, we kept an additional identifier indicating whether the cross-link originated from the decoy or target part of the fused entry.

## Subgrouping and FDR calculations

We devised only two subgroups for intra- and inter-dependent filtering to assure subgroups are sufficiently large to enable target-decoy competition (Fu and Qian, 2014) and to avoid overfitting to a specific dataset. In the case of intra-dependent subgrouping, we devised two subgroups: One context-rich group containing all inter-links where both the cross-linked proteins are additionally supported by intra-links and a second, context-poor, subgroup containing inter-links where at most one of the proteins contained intra-links. In the case of inter-dependent grouping, we also devised two subgroups. A context-rich subgroup containing the inter-links where each of the proteins in a PPI is supported by at least two inter-linked lysines involving different reactive sites on both ends and a second, context-poor, subgroup containing all other inter-link matches. Groupings were performed at the level of residue–residue pairs, and all residue–residue identifications without FDR control were used for subgrouping. Following subgrouping, we used previously established approaches for calculating posterior error probabilities (PEP) to estimate the probability distributions for correct and incorrect matches within individual subgroups (Cox and Mann, 2008). Briefly, we make two lists per group. One list containing the matches to the decoy part, and one list matching the target part. Then, we generated a histogram for each by gaussian kernel smoothing of the score distributions. We here used the lower (worse) negative decadic logarithmic XlinkX search scores reported for both sides of the residue pair as the overall score for the residue pair. The resulting distributions correspond to approximations to the probability densities for group-dependent correct and incorrect hits:

$$p(s, G) \text{ and } p(s, G | X = \text{false}),$$

where $X$ signifies a match to a correct or incorrect hit, $s$ is the negative decadic logarithmic XlinkX score and $G$ the placement of the identification into a context-rich or context-poor subgroup.

According to Bayes theorem, we calculated the PEP for each residue pair as given in Eq. (1).

$$p(X = \text{false} | s, G) = (p(s | X = \text{false}, G) * p(X = \text{false} | G)) / p(s | G). \tag{1}$$

Here, $p(s | X = \text{false}, G)$ describes the probability of observing a given score s, considering an incorrect match in subgroup G. $p(X = \text{false} | G)$ is the probability of an incorrect match in the subgroup G, calculated as the fraction of decoys in the subgroup, and $p(s, G)$ is the probability of observing the score s in subgroup G, considering both correct and incorrect matches.

The FDR was calculated by combining the PEP values from the different groups and the combined PEPs are sorted starting with the best (lowest). Then residue pairs are successively accepted until a desired FDR cutoff is reached according to Eq. (2)

$$\text{FDR} = (\text{TD} - \text{DD}) / \text{TT}, \tag{2}$$

where T denotes a target match, and D is a decoy match on either side of a cross-link, respectively.

The standard inter-intra separate strategy does not require PEP calculation since it does not depend on subgrouping. Instead, all inter-links are assigned to the same group, which was sorted by decreasing search scores. Again, we used the lower (worse) negative decadic logarithmic XlinkX search scores reported for both sides of

the residue pair as the overall score for the residue pair. The false-discovery rate for each entry was then calculated based on Eq. (2).

## Simulation

Two simulations were performed to estimate the proportion of false positives when employing two different subgrouping strategies (intra- and inter-dependent). In both cases, we simulated a distribution of false and correct cross-links based on parameters as obtained from the datasets (HEK293T, mito and virion). For intra-dependent subgrouping, we first simulated a distribution of correct matches. This was achieved by calculating the number of cross-links per target protein within the respective dataset, followed by fitting a Zipfian (Furusawa and Kaneko, 2003) distribution through the ranked target cross-link counts per protein. A Zipfian power-law distribution was used to model the correct cross-links, reflecting the nature of real-world protein interaction networks. To simulate false matches, we calculated the number of cross-links per decoy protein and fitted a Poisson distribution through the ranked decoy cross-link counts per protein. Using the Poisson distribution assumes that wrong matches occur randomly, without considering the possibility of strong systematic biases towards specific proteins, such as through misidentified modified peptides (Bogdanow et al, 2016). The fit parameters of both models were utilized to create probability distributions for true and false matches. Then we placed correct intra-links (corresponding in number to the target intra-link count), correct inter-links (corresponding in number to the target inter-link count) and incorrect inter-links (corresponding in number to the decoy inter-link count) according to the protein-dependent match probabilities for correct and false matches onto all proteins considered in the database search. Subsequently, the fraction of incorrect inter-links was evaluated in the subgroups of inter-links where (i) both linked proteins contained at least one intra-link and (ii) all others.

For simulating the frequency of false matches upon inter-dependent subgrouping, the simulation was adapted as follows: (i) intra-links were not considered, and (ii) inter-link matches were split into two parts, reflecting the possibility that each of the two cross-linked peptides could be wrongly matched (target-target, decoy-target, decoy-decoy). Again, a Poisson distribution was fitted to the empirically observed counts of decoy matches for all cross-links per PPI involving the most frequently linked decoy protein. To simulate correct matches, a Zipfian distribution was fitted for all cross-links per PPI involving the most frequently linked target protein. Incorrect and correct matches were then deposited on both sides of the inter-link according to the resulting probabilities for all possible entries in the database (as described in the previous paragraph). The frequency of incorrect matches was evaluated in the subgroups of inter-links where (i) both of the linked proteins contained at least one additional inter-link between the same proteins and (ii) all others.

## AF2 prediction and structure mapping

Hetero-dimers were predicted by AF2 multimer v2.2 (Evans et al, 2022) using the default protocol and sequence search methods. For each model, the pDockQ score was calculated (Bryant et al, 2022). Different model qualities were considered based on the pDockQ score. Mapping of cross-links on AF2-Multimer predictions was performed using the bio3d R package. Therefore, we extracted the C-alpha atom coordinates of inter-linked Lysines in the predicted zero-rank dimeric structure and calculated their distance in three-dimensional space.

## Data availability

Code related to FDR calculations and simulations and a R script for the context-sensitive FDR analysis of MSAnnika data is provided under https://github.com/Bogdanob/XLMS_decoyFusion/.

The source data of this paper are collected in the following database record: biostudies:S-SCDT-10_1038-S44320-024-00079-w.

## Peer review information

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

## Acknowledgements

The authors acknowledge Philip Lössl (Absea Biotechnology, Berlin) for editing and critically reviewing the manuscript. BB acknowledges funding from DFG grant BO 5917/1-1. CW, MR, and JR are supported by the European Research Council (ERC) Starting Grant (ERC-STG No. 949184). LM is funded by the Leibniz-Wettbewerb (K284/2019). The AF2 computations were enabled by resources provided by the Swedish National Infrastructure for Computing (SNIC) at NSC (Berzelius), partially funded by the Swedish Research Council through grant agreement no. Berzelius-2021-29 and SNIC 2022/5-282 (AE). The work was funded by Deutsche Forschungsgemeinschaft (DFG) project LI 3260/6-1.

## Author contributions

**Boris Bogdanow**: Conceptualization; Data curation; Formal analysis; Validation; Investigation; Visualization; Methodology; Writing—original draft; Writing—review and editing. **Max Ruwolt**: Formal analysis. **Julia Ruta**: Formal analysis. **Lars Mühlberg**: Formal analysis. **Cong Wang**: Formal analysis. **Wen-feng Zeng**: Formal analysis; Investigation. **Arne Elofsson**: Formal analysis; Funding acquisition; Investigation; Methodology. **Fan Liu**: Conceptualization; Supervision; Funding acquisition; Investigation; Methodology; Writing—original draft; Project administration; Writing—review and editing.

Source data underlying figure panels in this paper may have individual authorship assigned. Where available, figure panel/source data authorship is listed in the following database record: biostudies:S-SCDT-10_1038-S44320-024-00079-w.

## Funding

## Disclosure and competing interests statement

FL is a shareholder and advisory board member of Absea Biotechnology Ltd. and VantAI. The remaining authors declare no competing interests.

# Expanded View Figures

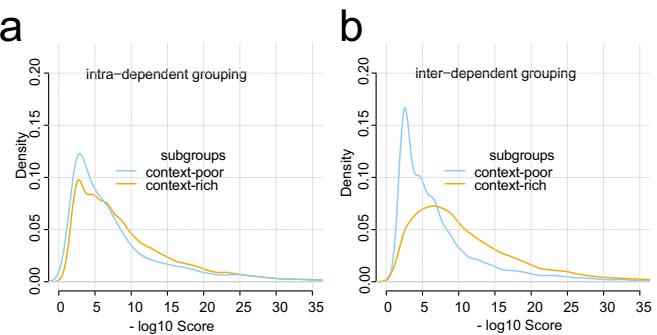

**Figure EV1.   Divergent error rates in context-dependent inter-link subgroups.**

(**A**, **B**) Score distributions of context-rich and context-poor groups upon intra-dependent subgrouping (**A**) or inter-dependent subgrouping (**B**) strategy in the "HEK293T" dataset.

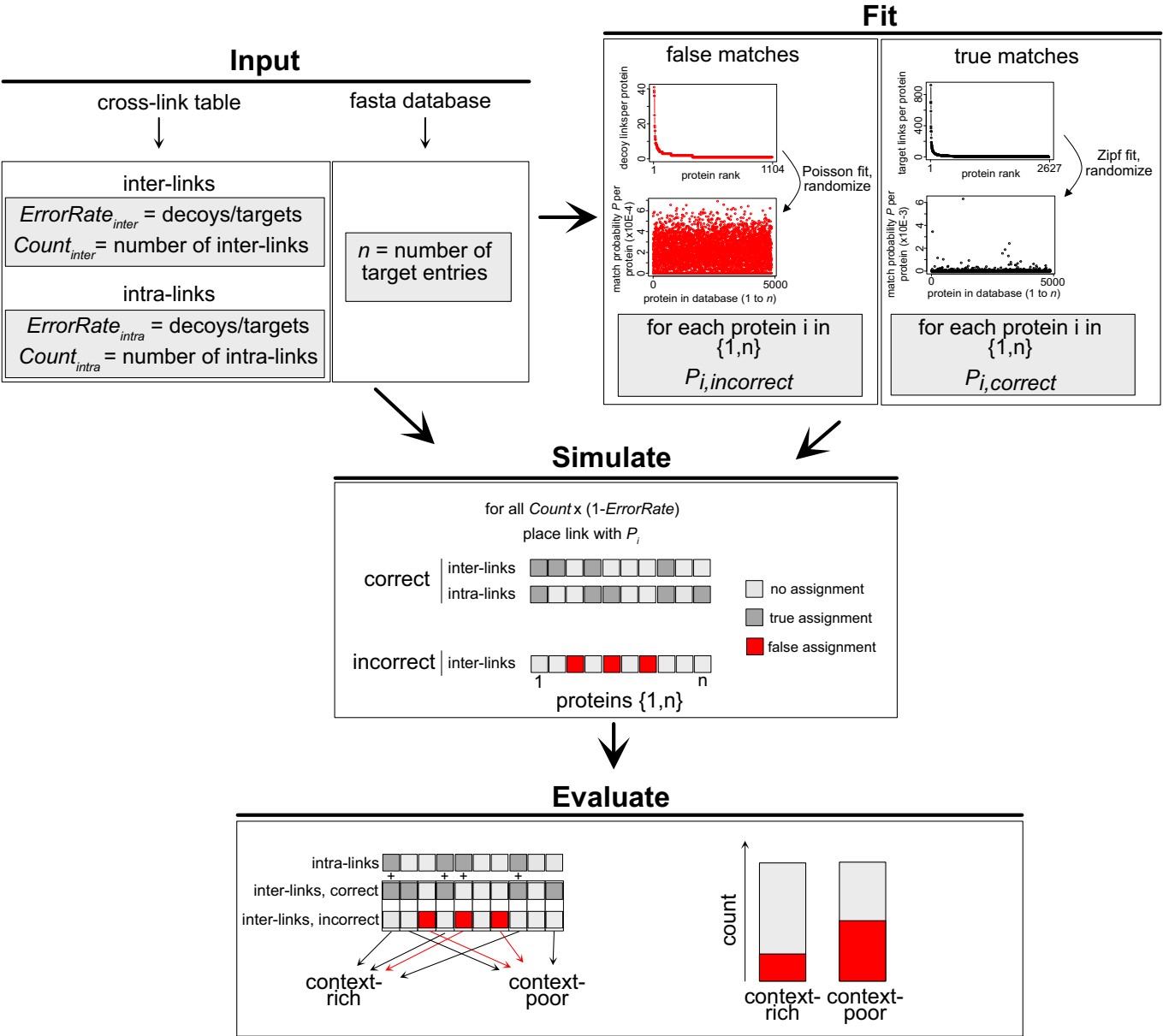

**Figure EV2.  Simulating false positives in XL-MS datasets.**

Workflow for simulations. Database size, the number of cross-links (inter and intra) and their decoy/target fractions were used as input. Protein-dependent probabilities for wrong matches were obtained from a Poisson fit through the decoy cross-link count per protein. Protein-dependent probabilities for correct matches were obtained from a Zipf fit through the target cross-link count per protein (Fit). Correct and Incorrect inter-links were deposited according to these probabilities on the proteins in the database. Incorrect intra-links were not considered. False and true positives were evaluated by grouping them into context-rich and context-poor groups, as schematically depicted for intra-dependent subgrouping.

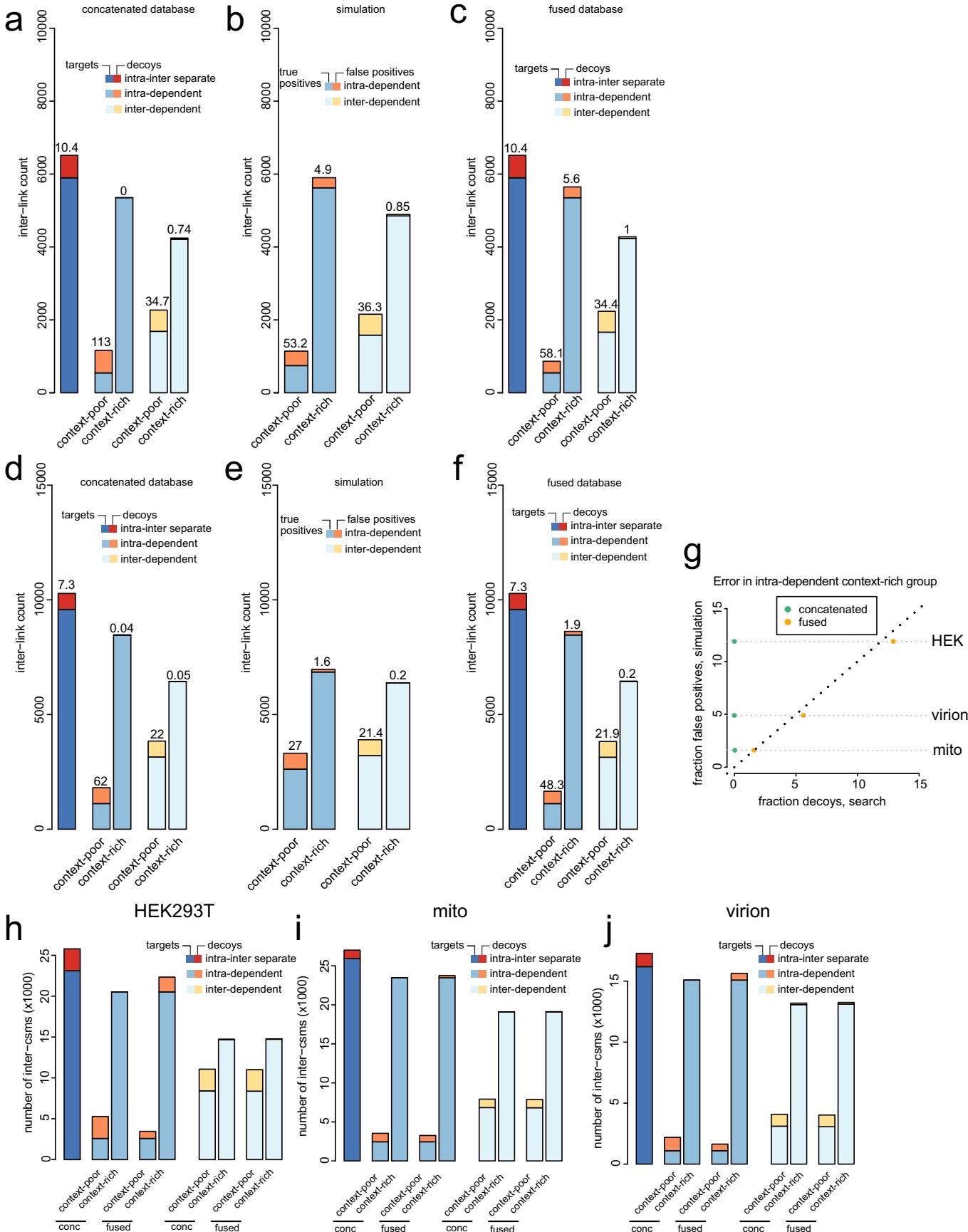

◄    **Figure EV3.   Target-decoy fusion agrees with simulated error rates upon context-sensitive subgrouping.**

(**A–F**) Distribution of inter-link matches to target or decoy entries in the "virion" (**A**, **C**) or "mito" (**D**, **F**) dataset upon intra-inter separate grouping or subgrouping of inter-link matches according to intra- and inter-dependent criteria using a concatenated (**A**, **D**) or fused (**C**, **F**) database. Simulation data on the distribution of false and true positives in context-rich and context-poor subgroups is given in panels (**B**) for "virion" and (**E**) for "mito" datasets. The number on top of the bars corresponds to the fraction of decoys to targets (**A**, **C**, **D**, **F**) or false positives to true positives (**B**, **E**) in percent. (**G**) Inter-link FDR in the context-rich, intra-dependent subgroup upon simulation, and upon fused or concatenated search for three biological datasets. The diagonal dotted line indicates a full agreement between simulation and search. (**H–J**) Distribution of inter-CSMs in different subgroups, as indicated for HEK293T, mito and virion datasets.

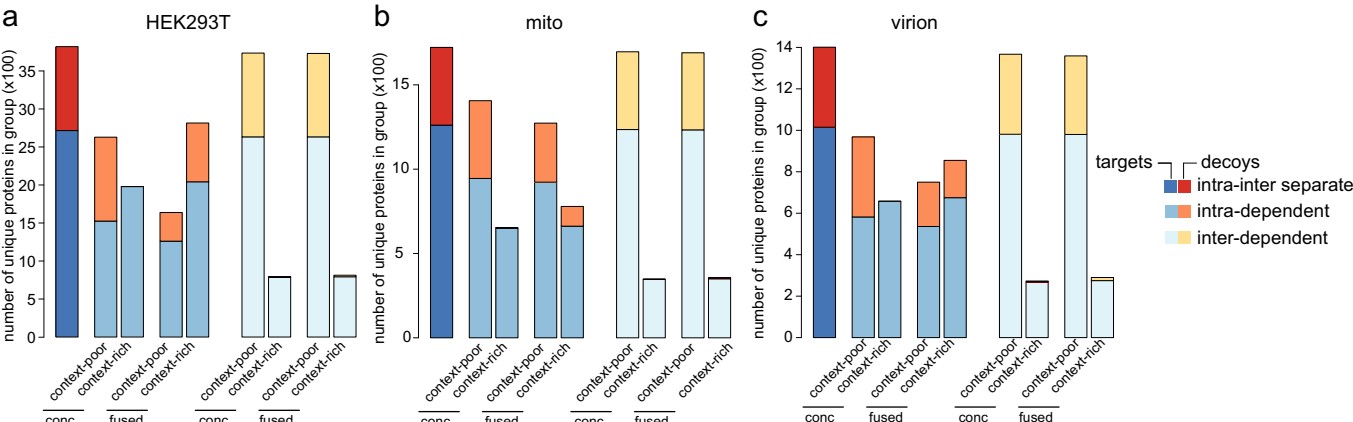

**Figure EV4. Unique Proteins in different subgroups.**

(A–C) Unique target or decoy proteins in subgroups from the HEK293T (**A**), mito (**B**) or virion (**C**) dataset upon intra-inter separate grouping or subgrouping of inter-link matches according to intra- and inter-dependent criteria using a concatenated or fused databases.

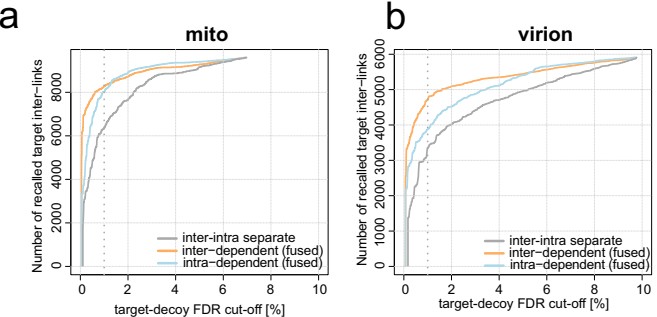

**Figure EV5. Improvement in statistical power through context-sensitive subgrouping.**

Recall of target inter-links as a function of the FDR cutoff comparing different data grouping strategies on a fused database for the (**A**) mito and (**B**) virion datasets.

