## [Peer Review File · Molecular Systems Biology]

Redesigning error control in XL-MS enables more robust&sensitive protein-protein interaction studies

Boris Bogdanow, Max Ruwolt, Julia Ruta, Lars Muehlberg, Cong Wang, Wen-Feng Zeng, Arne Elofsson, and Fan Liu

Corresponding author(s): Fan Liu (FLiu@fmp-berlin.de)

Review Timeline:

Submission Date:	27th May 24
Editorial Decision:	3rd Jul 24
Revision Received:	30th Sep 24
Editorial Decision:	8th Nov 24
Revision Received:	20th Nov 24
Accepted:	21st Nov 24

Editor: Jingyi Hou

Transaction Report:

3rd Jul 2024

Manuscript Number: MSB-2024-12447

Title: Redesigning error control in XL-MS enables more robust&sensitive protein-protein interaction studies

Author: Boris Bogdanow

Max Ruwolt

Julia Ruta

Lars Muehlberg

Cong Wang

Wen-Feng Zeng

Arne Elofsson

Fan Liu

Dear Prof Liu,

Thank you for submitting your work to Molecular Systems Biology. We have now heard back from the three reviewers who agreed to evaluate your manuscript. As you will see from the reports below, the reviewers acknowledge the interest of the study. They raise, however, a series of concerns, which we would ask you to address in a major revision.

The reviewers' recommendations are relatively clear, so there is no need to reiterate the points listed below. In particular, both Reviewers #1 and #3 mentioned a recent study by Fischer and Rappsilber that proposed a similar strategy. In light of Reviewer #1's comment, we would ask you to cite this study. While a direct comparison and discussion of both methods suggested by this reviewer would be welcome, it is not required for the acceptance of the manuscript.

All other issues raised by the reviewers need to be satisfactorily addressed as well. As you may already know, our editorial policy allows in principle a single round of major revision, and it is therefore essential to provide responses to the reviewers' comments that are as complete as possible.

On a more editorial level, we would ask you to address the following issues:

- Please provide a .docx formatted version of the manuscript text (including legends for main figures, EV figures and tables). Please make sure that the changes are highlighted to be clearly visible.

- Please provide individual production quality figure files as .eps, .tif, .jpg (one file per figure).

-Please provide a .docx formatted letter INCLUDING the reviewers' reports and your detailed point-by-point responses to their comments. As part of the EMBO Press transparent editorial process, the point-by-point response is part of the Review Process File (RPF), which will be published alongside your paper.

-Please note that all corresponding authors are required to supply an ORCID ID for their name upon submission of a revised manuscript.

-We replaced Supplementary Information with Expanded View (EV) Figures and Tables that are collapsible/expandable online (see examples in <http://msb.embopress.org/content/11/6/812>). A maximum of 5 EV Figures can be typeset. EV Figures should be cited as 'Figure EV1, Figure EV2' etc... in the text and their respective legends should be included in the main text after the legends of regular figures.

Additional Tables/Datasets should be labeled and referred to as Table EV1, Dataset EV1, etc. Legends have to be provided in a separate tab in case of .xls files. Alternatively, the legend can be supplied as a separate text file (README) and zipped together with the Table/Dataset file.

For the figures and tables that you do NOT wish to display as Expanded View figures, they should be bundled together with their legends in a single PDF file called *Appendix*, which should start with a short Table of Content. Each legend should be below the corresponding Figure/Table in the Appendix. Appendix figures and tables should be referred to in the main text as: "Appendix Figure S1, Appendix Figure S2, Appendix Table S1" etc. See detailed instructions regarding expanded view here: <https://www.embopress.org/page/journal/17444292/authorguide#expandedview>.

-Before submitting your revision, primary datasets (and computer code, where appropriate) produced in this study need to be deposited in an appropriate public database (see<http://msb.embopress.org/authorguide> - dataavailability <https://www.embopress.org/page/journal/17444292/authorguide#dataavailability>).

The accession numbers and database should be listed in a formal "Data Availability" section (placed after Materials & Method) that follows the model below (see also <https://www.embopress.org/page/journal/17444292/authorguide#dataavailability>). Please note that the Data Availability Section is restricted to new primary data that are part of this study.

Data availability

-At EMBO Press we ask authors to provide source data for the main figures. Our source data coordinator will contact you to discuss which figure panels we would need source data for and will also provide you with helpful tips on how to upload and organize the files.

- Our journal encourages inclusion of *data citations in the reference list* to directly cite datasets that were re-used and obtained from public databases. Data citations in the article text are distinct from normal bibliographical citations and should directly link to the database records from which the data can be accessed. In the main text, data citations are formatted as follows: "Data ref: Smith et al, 2001". In the Reference list, data citations must be labeled with "[DATASET]". A data reference must provide the database name, accession number/identifiers and a resolvable link to the landing page from which the data can be accessed at the end of the reference. Further instructions are available at .

- We updated our journal's competing interests policy in January 2022 and request authors to consider both actual and perceived competing interests. Please review the policy <https://www.embopress.org/competing-interests> and update your competing interests if necessary.

Please use the heading "Disclosure statement and competing interests".

- All Materials and Methods need to be described in the main text using our 'Structured Methods' format, which is required for all research articles. According to this format, the Methods section includes a Reagents and Tools Table (listing key reagents, experimental models, software and relevant equipment and including their sources and relevant identifiers) followed by a Methods and Protocols section describing the methods using a step-by-step protocol format. The aim is to facilitate adoption of the methodologies across labs. More information on how to adhere to this format as well as a downloadable template (.docx) for the Reagents and Tools Table can be found in our author guidelines:

<https://www.embopress.org/page/journal/17444292/authorguide#structuredmethods>.

-Regarding data quantification:

Please ensure to specify the name of the statistical test used to generate error bars and P values, the number (n) of independent experiments (please specify technical or biological replicates) underlying each data point and the test used to calculate p-values in each figure legend. Discussion of statistical methodology can be reported in the materials and methods section, but figure legends should contain a basic description of n, P and the test applied.

Graphs must include a description of the bars and the error bars (s.d., s.e.m.).

- Please provide a "standfirst text" summarizing the study in one or two sentences (approximately 250 characters, including space), three to four "bullet points" highlighting the main findings and a "synopsis image" (550px width and 400-600 px height, PNG format) to highlight the paper on our homepage.

Here are a couple of examples:

<https://www.embopress.org/doi/10.15252/msb.20199356>

<https://www.embopress.org/doi/10.15252/msb.20209475>

<https://www.embopress.org/doi/10.15252/msb.209495>

When you resubmit your manuscript, please download our CHECKLIST (<https://www.embopress.org/pb-assets/embosite/EMBO%20Press%20Author%20Checklist-1642513524327.xlsx>) and include the completed form in your submission.

Please note that the Author Checklist will be published alongside the paper as part of the transparent process (<https://www.embopress.org/page/journal/17444292/authorguide#transparentprocess>).

If you feel you can satisfactorily deal with these points and those listed by the referees, you may wish to submit a revised version

of your manuscript. Please attach a covering letter giving details of the way in which you have handled each of the points raised by the referees. A revised manuscript will be once again subject to review and you probably understand that we can give you no guarantee at this stage that the eventual outcome will be favorable.

I look forward to receiving your manuscript soon.

Sincerely,
Jingyi

Jingyi Hou, PhD
Scientific Editor
Molecular Systems Biology

We realize that it is difficult to revise to a specific deadline. In the interest of protecting the conceptual advance provided by the work, we recommend a revision within 3 months (1st Oct 2024). Please discuss the revision progress ahead of this time with the editor if you require more time to complete the revisions. Use the link below to submit your revision:

IMPORTANT: When you send your revision, we will require the following items:

1. the manuscript text in LaTeX, RTF or MS Word format
2. a letter with a detailed description of the changes made in response to the referees. Please specify clearly the exact places in the text (pages and paragraphs) where each change has been made in response to each specific comment given
3. three to four 'bullet points' highlighting the main findings of your study
4. a short 'blurb' text summarizing in two sentences the study (max. 250 characters)
5. a 'thumbnail image' (550px width and max 400px height, Illustrator, PowerPoint or jpeg format), which can be used as 'visual title' for the synopsis section of your paper.

6. Please include an author contributions statement after the Acknowledgements section (see

<https://www.embopress.org/page/journal/17444292/authorguide>)

7. Please complete the CHECKLIST available at (<https://bit.ly/EMBOPressAuthorChecklist>).

Please note that the Author Checklist will be published alongside the paper as part of the transparent process

(<https://www.embopress.org/page/journal/17444292/authorguide#transparentprocess>).

See also figure legend guidelines: <https://www.embopress.org/page/journal/17444292/authorguide#figureformat>

9. Please note that corresponding authors are required to supply an ORCID ID for their name upon submission of a revised manuscript (EMBO Press signed a joint statement to encourage ORCID adoption).

(<https://www.embopress.org/page/journal/17444292/authorguide#editorialprocess>)

Currently, our records indicate that the ORCID for your account is 0000-0002-2358-549X.

Link Not Available

11. Include a Reagents and Tools Table as part of the Methods section, which can be downloaded from our author guidelines (<https://www.embopress.org/page/journal/17444292/authorguide#structuredmethods>)

*** PLEASE NOTE *** As part of the EMBO Press transparent editorial process initiative (see our Editorial at <https://dx.doi.org/10.1038/msb.2010.72>), Molecular Systems Biology publishes online a Review Process File with each accepted manuscripts. This file will be published in conjunction with your paper and will include the anonymous referee reports, your point-by-point response and all pertinent correspondence relating to the manuscript. If you do NOT want this File to be published, please inform the editorial office at msb@embo.org within 14 days upon receipt of the present letter.

Reviewer #1:

Liu and co-workers present a new approach for error rate control in large-scale cross-linking mass spectrometry (XL-MS). While proteome-wide applications of XL-MS have increased in the last few years, it remains challenging to maximize the number of confidently identified protein-protein interactions (PPIs), irrespective of the specific experimental design. This is due to the vastly increased search space when searching large sequence databases. Different computational approaches have been designed to address this. By now, it is fairly established that evaluating intra- and inter-protein cross-links with the same target/decoy (T/D) model leads to a (sometimes severe) underestimation of the false discovery rate (FDR) at the PPI level. Strategies that take into account "circumstantial evidence", for example, considering only inter-protein cross-links on proteins for which intra-protein links have been observed, risk violating certain assumptions of the T/D method, as is the case for a method recently described by Stengel and co-workers.

In the present work, Bogdanow et al. introduce a refined concept for search space restriction based on a so-called "fusion" T/D strategy, which has previously been proposed for the analysis of regular proteomics data. Here, the authors first use an in-house generated dataset to confirm that the joint intra/inter search leads to an inflated FDR at the inter-protein cross-link level, while separating the two cross-link types is potentially overly conservative, leading to a substantial increase of false negatives among inter-protein cross-links for this "synthetic" dataset with known ground truth. The authors then proceed with more refined filtering approaches. The "mi-filter" strategy proposed by the Stengel group does not adequately control inter-protein group FDR. The new strategy proposed here relies on the concurrent presence of other inter-protein cross-links between the same proteins ("context-rich"), which yields a more balanced FDR control. Finally, the fusion concept, whereby each target sequence appended with its decoy sequence as one entry is applied, which is reported to further improve the accuracy of FDR control. This strategy is then applied to several published data sets, and additionally identified inter-protein cross-links are evaluated with predicted models of complexes to add confidence to the newly "accepted" hits.

The manuscript addresses a relevant topic, namely the FDR control in large-scale cross-linking datasets at the inter-protein cross-link or PPI level, which is a hotly debated topic. Therefore, the manuscript is clearly of interest to many readers of MSB. However, I suggest that the authors address two major points in a revised version of the manuscript before publication. I have also a few minor comments summarized below.

Major comments:

First, I am sure that the authors are aware of a recent pre-print of Fischer and Rappsilber, which also critiques the mi-filter strategy of the Stengel group (Chen et al.), and where another "workaround" is proposed. Although the necessity of citing preprints can be debated, in the context of this work it is important to discuss both papers. I suggest that the authors compare the two approaches accordingly.

Second, the authors claim that their fusion database method is a "universal strategy", however, it must obviously depend on a certain depth of data so that other cross-links supporting the same PPI can be identified in sufficient numbers. This raises several questions. For example, what is the lower limit of dataset coverage for this strategy to be superior to a more classical strategy, e.g. the independent T/D search for intra- and inter-protein links without further filters? The authors should be able to assess this with their simulations. How does the authors' method compare with a simple filter that would only consider interactions with at least two PSMs supporting a PPI?

Minor comments:

Method section, search parameters: Which cross-linking sites were allowed, only Lys? This is implied from some other statements in the text, but it is not explicitly mentioned.

Method section, definition of "context-rich" interactions: Do the additional cross-links need to involve other reactive sites on both ends of the cross-link or just one? This is not really clear from the method section but is hinted at in the legend to Figure 2.

The legend to Figure 1a says "... proteins were denatured and cross-linked in groups of eight ...", citing Ref. 17. When checking Ref. 17, it reads "The dissolved proteins were mixed in pairs of two proteins in all possible combinations within one interaction group. They were incubated for 20 min at 50 {degree sign}C to induce interactions in vitro. 0.2- 1 mM DSSO cross-linking reagent was added to the groups and incubated at room temperature for 30 min." Please clarify.

Reference #1 needs to be cleaned up.

Plots in Fig. 4 and Extended Data Fig. 4 use different color schemes, which is a bit confusing.

Reviewer #2:

Summary:

This paper explores the use of different strategies to assess and control false discovery rate (FDR) in cross-linking mass spectrometry (XL-MS) studies, and proposes a new strategy for higher quality XL-MS datasets. As erroneous identifications usually present as inter-protein cross-links, this is of high interest for large scale XL-MS studies where high confidence protein-protein interaction mapping is often the goal. The authors proposed a new strategy for FDR control which combined two main steps: (1) the use of a target-decoy fusion database (as opposed to the usually used concatenated target-decoy databases), and (2) the grouping of cross-linked spectra populations by level of parallel observations as well as by cross-link information type (intra- vs inter-link) BEFORE the FDR control takes place. This enabled effective control of false positives whilst substantially decreasing the rate of false negatives, producing more high quality cross-links that represent bona-fide PPIs. The authors assessed this FDR strategy (alongside several others suggested and used by the field) using a unique ground-truth dataset (described in another paper by the same research group), a synthetic dataset, and real XL-MS datasets from different organisms. They also investigated how its use improved interactome coverage and the interpretation of structural models of protein-protein interactions.

Overall, the paper is well written. The findings presented by the authors are exciting and present a key conceptual and technical finding that would be of very high interest and utility for the XL-MS community.

Minor points:

If implemented by popular XL-MS search engines, the strategy should improve the sensitivity and accuracy of future XL-MS studies. The authors show that this strategy works on posthoc FDR analyses from cross-link data produced by XlinkX, which does not have the fusion/context-subgrouping strategy in-built. Do these findings generalise to the identifications produced from other search engines such as pLink2, MSAnnika, MeroX, etc? How much better does the FDR control get beyond what is performed by default in these other XL-MS programs?

Although not necessary, one way to greatly improve the study's direct usefulness for the field would be to include a tool (perhaps even just an R script) that can (1) generate the fused target-decoy FASTA files, and (2) perform their subgrouped post-hoc FDR analysis on the output of commonly used search engines. Such a tool should report the error at all levels of biological redundancy (spectral match, peptide pair, residue pair, protein pair).

Given its importance for the paper, the explanation of the benefits of using a target-decoy fusion over a concatenated target-decoy database could be explained better. After some reading, I now understand the motivations for its use, but just a few more sentences explicitly stating why it would better model the error for the subgroup FDR strategy would greatly increase the readability of the paper. On a similar note, the conceptual explanation of how the synthetic dataset was generated could be clearer.

Reviewer #3:

This manuscript is a timely addition to advancing crosslinking MS analysis beyond studying individual protein complexes toward whole proteomes. The authors address the challenge of balancing sensitivity and specificity in identifying protein-protein interactions (PPIs) by developing a target-decoy fusion strategy that allows context-dependent data filtering. Before this study a few groups have suggested ways to filter on protein groups and have shown impressive results, however, as the authors discuss, filtering on the protein level using a concatenated target-decoy database can cause the link between targets and decoys to be lost. To address this the authors have used a fused-decoy strategy like the one (recently preprinted) by Fischer and Rappsilber. Together these manuscripts demonstrate the current way to subgroup crosslinking MS data on the protein level and I enthusiastically support this manuscript for publication.

Comments:

From reading the manuscript I think that the main gain from the fused-decoy approach is that crosslinks can now be filtered at the protein level. This made me consider whether the context-dependent approach suggested by the authors is the optimal use of information in the real biological datasets. I would like the authors to try to look at the effects of protein abundance information instead to see how it compares.

Typically, before crosslink search on a proteome-scale sample it is standard practice to filter the proteins searched by an abundance cutoff from standard proteomics. This dramatically helps boost matches as crosslinked peptides tend to only be found in the most abundant proteins. My concern with that approach has always been that linear (and linear-modified) peptides from proteins that aren't included in the search can be matched as decoys or falsely matched as crosslinks. I would like to see if

the authors' fused-decoy approach can allow filtering on protein abundances *after* search. I would guess that the chances of an identified crosslink being 'real' increase at higher protein abundances.

My thinking is that protein intensity filtering is already happening with the context-dependent approach anyway, with proteins counted as 'context-dependent' being much more abundant than those in the other group. It would be useful if the authors checked this in the HEK dataset and any others.

The size of each subgroup is important information if there is to be an accurate estimation of FDR based on the decoys remaining. The percentages and ratios in the figures are important, but it is also useful for the reader to know the number of CSMs, residue pairs, and proteins in each subgroup.

The methods are missing two key pieces of information that would allow interpretation of the results, 1- that the data was acquired as MS2, and 2- how XlinkX handles merging from CSMs up through the levels to crosslinked residue pairs. I assume only top-scored CSMs are used for each spectrum throughout. Also, I assume when crosslinks are discussed in the manuscript that refers to 'unique residue pairs'?

I do not understand the step of merging of the subgroups to generate a final combined list. If each subgroup had 1% FDR and they are truly separate groups (no overlap of matched spectra) why can't they just be merged and the FDR percentage maintained. Combining the results post 5% FDR should still give you 5% FDR in the combined list. If you get different results after combining something is off.

Another issue is that if one uses false discovery rates of each match in the merging you only have the resolution of the numbers of decoys you have remaining. If the overall numbers are quite small you get large jumps in FDR between matches.

Minor comments

- Please use HEK293T instead of HEK throughout.

- The figures could be improved with better labeling. Particularly the numbers on and inside bars.

- With the controlled dataset the authors have, have they considered other ways of filtering such as those discussed in the introduction, such as different sizes of crosslinked peptides?

- The HEK dataset give 12,216 targets and only 2003 decoys without FDR filtering. This seems low, but I assume only spectra with DSSO doublets are reported as matches?

Point-by-point response to the reviewer's comments

We thank all three reviewers for reading our manuscript and for the constructive feedback and comments! We were particularly pleased to read that the topic is “clearly of interest to many readers” (reviewer #1), that our findings are “exciting and a key conceptual and technical finding” (reviewer #2) and that this manuscript is “enthusiastically support[ed] [...] for publication” (reviewer #3).

Issues raised by the reviewers were to

(i) reference the pre-print by Fischer and Rappsilber, 2023, now published in MSB, and compare ours and their work (reviewer #1).

(ii) investigate the potential of the workflow across different scales of coverage, the use of other subgrouping criteria and search engines (reviewer #1-3).

We have now carefully addressed these and other concerns and provide a new, revised version of the manuscript that we feel is ready for publication in Molecular Systems Biology.

Reviewer #1:

Liu and co-workers present a new approach for error rate control in large-scale cross-linking mass spectrometry (XL-MS). While proteome-wide applications of XL-MS have increased in the last few years, it remains challenging to maximize the number of confidently identified protein-protein interactions (PPIs), irrespective of the specific experimental design. This is due to the vastly increased search space when searching large sequence databases. Different computational approaches have been designed to address this. By now, it is fairly established that evaluating intra- and inter-protein cross-links with the same target/decoy (T/D) model leads to a (sometimes severe) underestimation of the false discovery rate (FDR) at the PPI level. Strategies that take into account "circumstantial evidence", for example, considering only inter-protein cross-links on proteins for which intra-protein links have been observed, risk violating certain assumptions of the T/D method, as is the case for a method recently described by Stengel and co-workers.

In the present work, Bogdanow et al. introduce a refined concept for search space restriction based on a so-called "fusion" T/D strategy, which has previously been proposed for the analysis of regular proteomics data. Here, the authors first use an in-house generated dataset to confirm that the joint intra/inter search leads to an inflated FDR at the inter-protein cross-link level, while separating the two cross-link types is potentially overly conservative, leading to a substantial increase of false negatives among inter-protein cross-links for this "synthetic" dataset with known ground truth. The authors then proceed with more refined filtering approaches. The "mi-filter" strategy proposed by the Stengel group does not adequately control inter-protein group FDR. The new strategy proposed here relies on the concurrent presence of other inter-protein cross-links between the same proteins ("context-rich"), which yields a more balanced FDR

control. Finally, the fusion concept, whereby each target sequence appended with its decoy sequence as one entry is applied, which is reported to further improve the accuracy of FDR control. This strategy is then applied to several published data sets, and additionally identified inter-protein cross-links are evaluated with predicted models of complexes to add confidence to the newly "accepted" hits.

The manuscript addresses a relevant topic, namely the FDR control in large-scale cross-linking datasets at the inter-protein cross-link or PPI level, which is a hotly debated topic. Therefore, the manuscript is clearly of interest to many readers of MSB. However, I suggest that the authors address two major points in a revised version of the manuscript before publication. I have also a few minor comments summarized below.

We thank the reviewer for the positive comments!

Major comments:

First, I am sure that the authors are aware of a recent pre-print of Fischer and Rappsilber, which also critiques the mi-filter strategy of the Stengel group (Chen et al.), and where another "workaround" is proposed. Although the necessity of citing preprints can be debated, in the context of this work it is important to discuss both papers. I suggest that the authors compare the two approaches accordingly.

We are aware of this pre-print and fully agree with the reviewer that this represents an additional reference relevant to the context of this paper and therefore of interest for the readers.

*While we have now added this citation (see below), we would like to point out that our study (<https://doi.org/10.1101/2023.07.19.549678>, online July 19, 2023 on bioRxiv) was pre-printed >4 months earlier than the work by Fischer and Rappsilber (<https://doi.org/10.1101/2023.11.28.568978>, online Nov 28, 2023). We were surprised to see that the study by Fischer and Rappsilber, now published in *Mol Syst Biol*, neither references our earlier pre-print nor prior work by others in shotgun proteomics who developed decoy-fusion strategies (Zhang et al., 2012, MCP, referenced in our manuscript), applied and fine-tuned them (Savitski et al., 2015, MCP, referenced in our manuscript).*

We appreciate the reviewer's suggestion for a direct comparison to the Fischer & Rappsilber work. However, there are important differences between this and our approach that limit the potential for an accurate comparison:

- *First, Fischer and Rappsilber exclusively focused on restoring error control of the mi-filter. In our study rescuing error control is only one aspect next to presenting a viable strategy on how sensitivity can be increased. Our work therefore goes much beyond demonstrating problems of other workflows but instead presents a more positive view on*

how the overall problem can be leveraged for more robust & sensitive protein-protein interaction studies.

- Second, a filter as applied by Fischer and Rappsilber is different to subgrouping (our approach). While the filter discards matches with lower confidence, our approach retains them but requires higher cut-offs to accept them in the final list of matches. Thus, filtering reduces information content (“filter loss”), whereas subgrouping accounts for the relative differences of error rates within the subgroups without loss. Compared to a subgrouping approach, a filtering approach results in overall fewer inter-links and a loss of sensitivity in all biological datasets included in our study (see Reviewer Figure 1 below). Therefore, the subgrouping approach is conceptually superior as it retains the overall information content of the dataset and only gives different weightings to matches separately for subgroups.

Reviewer Figure 1. Comparison of the intra-dependent subgrouping approach to the intra-dependent filter for three datasets (left: HEK293T, middle: mito and right: virion). Recall of target inter-links as a function of the applied inter-link FDR. In the intra-dependent filter, all inter-links in the context-poor subgroup were removed prior to FDR analyses (e.g. those inter-links, where the corresponding proteins do not contain intra-links). Note that this results in a “filter loss” exemplified by lower maximum counts of recalled inter-links. At best, the filtering strategy performs identical to the herein presented subgrouping method. The vertical line indicates FDR cut-off at 1 %.

- Third, the approach by Fischer and Rappsilber proposes using the “boost” option in xiFDR/xiSearch for best performance. This “boost” option yields improvements that outweigh any benefits of the implemented corrected version of the mi-filter. “Boost” takes into account that cross-links can be identified on different levels, namely cross-link spectrum matches (CSMs) → unique residue pairs (ResPairs) → protein-protein interactions (PPIs), and tries to maximize recalled IDs at a higher level by varying lower-level FDR cut-offs. However, the “boost” option is not necessarily a good idea when accurate error control is desired. When testing “Boost”-enabled XiFDR/XiSearch on the ground-truth XL-MS dataset (Clasen & Ruwolt et al., accepted in Nature Methods, see figure below), the empirical PPI-FDR is more than twice as high as the desired FDR cut-off (11.5 % instead of 5 %). XiFDR/XiSearch only encounters this FDR inflation when

“boost” is enabled. This hints at a problem in the idea/implementation of “boost” and cautions against a direct comparison of our strategy against the Fischer & Rappsilber approach since the effects of their filter are smaller than the “boost effects” and the two cannot be disentangled.

Reviewer Figure 2. Benchmarking of Scout against xiSEARCH/xiFDR. (A) Inter-protein CSM, ResPairs and PPI identifications at 5% PPI-FDR and empirically determined FDR using Scout and xiSEARCH (both with default parameters, xiFDR with boost between proteins) on a subset of the benchmarking dataset using a 540-protein database and KSTY as possible reaction sites for the cross-linking reagent. Note that Scout is novel software developed in the Clasen and Ruwolt et al. paper. This figure is identical to supplementary figure 2 from Clasen and Ruwolt et al., 2024, Nat Methods (accepted).

The above points illustrate that a direct comparison between our approach and the one from the Rappsilber lab is not trivial and requires a lot of contextualization. Even the above discussion is not as exhaustive as one would expect for a published article, because – for example – we can only speculate as to why the “boost” function seems to inflate the PPI-FDR. We hope the reviewer will agree that such a lengthy and yet not fully conclusive comparison is not well suited within this manuscript and may distract from the main findings. Therefore, we prefer to keep this discussion in the point-by-point response, which will still be published alongside the manuscript.

Action taken:

We cite the Fischer and Rappsilber work, now published in MSB, in the Discussion section as follows:

“To solve this issue, we adapted the decoy fusion database search design which was firstly introduced in shotgun proteomics workflows (Savitski et al, 2015; Zhang et al, 2012) and demonstrated its capabilities in restoring target decoy symmetry and allowing better control of error rates (Figure 3d,e). This is consistent with a similar strategy by the Rappsilber lab to restore error control of the mi-filter (Fischer & Rappsilber, 2024), which was preprinted and published after the preprint of this article appeared online.”

Second, the authors claim that their fusion database method is a "universal strategy", however,

it must obviously depend on a certain depth of data so that other cross-links supporting the same PPI can be identified in sufficient numbers. This raises several questions. For example, what is the lower limit of dataset coverage for this strategy to be superior to a more classical strategy, e.g. the independent T/D search for intra- and inter-protein links without further filters? The authors should be able to assess this with their simulations.

This is an excellent idea! We first attempted to address the robustness of our approach using a simulation. We simulated true and false inter-links and whether they fall into context-rich and context-poor subgroups upon inter-dependent subgrouping. We performed this analysis by considering varying true inter-link counts 120 - 12,000 (in increments of 120) and false inter-links (set to 20 % of the true inter-link count). True inter-links were deposited on proteins following a power-law distribution and false inter-links following a random distribution, as described in the method section. Then, we counted the number of true inter-links and the fraction of false inter-links in context-rich and context-poor subgroups (see Reviewer Figure 2, below).

*Trivially, we found that including overall fewer inter-links in the simulation leads to decreases in the context-poor and context-rich inter-links. Consistent with the reviewer's reasoning, we observed that the number of true inter-links in the context-rich subgroup decreased faster than the number of true inter-links in the context-poor subgroup (**Figure Reviewer 2, left**). However, the overall fraction of false matches in the context-rich subgroup remained stable at very low levels, consistent with our data on HEK293T, mito and virion datasets (**Figure Reviewer 2, right**). This analysis suggests that inter-dependent subgrouping may be beneficial until very low depth of data, albeit likely with less pronounced benefits.*

Reviewer Figure 2. Simulated inter-link count (**left**) and fraction of false inter-links (**right**) in context-rich and context-poor subgroups depending on the number of inter-links in the simulation. (Simulation parameters: true interlinks: 120-12,000, step size: 120 interlinks, overall probability of false inter-link match: 0.2, database size: 4,860 entries)

To see if this conclusion from the modeling holds in a real-world dataset, we assessed the robustness of our approach on the HEK293T dataset, dependent on the dataset depth.

Therefore, we randomly selected fractions of the ResPairs from the original dataset and evaluated the performance of the standard intra-inter separate or the two context-sensitive strategies (**Reviewer Figure 3**). In all cases, context-sensitive strategies performed better than the standard inter-intra separate strategy for identifying a higher number of inter-links. This holds true even if only 10% of the original data are included (representing 503 inter-links in the standard search), although with smaller differences between the strategies, consistent with the simulations (see above).

Reviewer Figure 3. Comparison of FDR-strategies across different dataset depths. Residue pairs from HEK293T (no FDR applied) cell dataset were randomly selected in different proportions (10% to 100%). Following this, subgrouping and FDR calculations were performed and the number of recalled target inter-links at 1% target-decoy FDR is depicted. Overall similar observations with respect to scale and benefit were made with the two other biological datasets.

We conclude that the simulations and the artificial reductions in dataset depth suggest that inter-dependent grouping can be beneficial down to low dataset depths. The reason for this is that no matter how small the context-rich subgroup is (for instance when an XL-MS dataset has very little coverage), there will always be some lower-scored links rescued by categorization in context-rich subgroup. In the extreme case, if no context-rich inter-links are identified, then the context-rich group collapses and all inter-links end up in the context-poor group. Then, FDR is computed exclusively within the context-poor group and the results will be identical to the standard intra-inter separate strategy. Therefore, the subgrouping strategy is universally suitable and beneficial in many, if not most, situations to increase coverage.

How does the authors' method compare with a simple filter that would only consider interactions with at least two PSMs supporting a PPI?

Typically, a filter as suggested by the reviewer is performed to increase the confidence of the reported matches after FDR control has been performed. The hope is that the filter affects false hits stronger than true hits so that the overall confidence of the reported set of identifications is

increased. Trivially, such a filter will report fewer inter-links and PPIs compared to the standard approach, although the confidence of the matches may be increased.

If the reviewer asks about the effectiveness of this simple PSM filter prior to FDR control, it is important that it is performed in a fused database design to rescue the decoy complement, which are overall sparser populated by cross-links than the targets.

To evaluate the proposed strategy, we turned to our HEK293T dataset, removed all ResPairs where only one CSM matched to the PPI (“inter-intra separate with 2PSM per PPI prefilter”), and counted the number of inter-links. We compared this strategy to inter-intra separate and the two subgrouping strategies (see Reviewer Figure 4, below) on the HEK293T dataset and found that the suggested strategy performed similar to the intra-inter separate (standard) and intra-dependent grouping strategies. Still, the inter-dependent grouping strategy performs decisively better than the suggested filtering strategy.

Reviewer Figure 4. Comparison of the suggested 2PSM per PPI pre-filter strategy (dotted line) to the strategies investigated in this manuscript (solid lines) for recall of inter ResPair identifications across various FDR thresholds.

Additionally, our approach is not a simple filter (such as the *mi*-filter from Stengel and co-workers or the *ec*-filter by Fischer and Rappsilber, also see our response to the first comment of this reviewer) but rather a new FDR control strategy. A filter discards matches deemed irrelevant or uncertain and thus necessarily comes with a loss of information. In contrast, our subgrouping approach considers different error likelihoods of context-poor and context-rich matches. Thus, all information content is included in our statistical framework which leads to the increases in sensitivity.

Minor comments:

Method section, search parameters: Which cross-linking sites were allowed, only Lys? This is implied from some other statements in the text, but it is not explicitly mentioned.

Yes, this is correct. We added an additional statement to the Methods section.

Method section, definition of "context-rich" interactions: Do the additional cross-links need to involve other reactive sites on both ends of the cross-link or just one? This is not really clear from the method section but is hinted at in the legend to Figure 2.

Yes, the reviewer is correct. The cross-links need to involve other reactive sites on both ends of the cross-links. This was not explicitly mentioned in the manuscript and is now added.

Action taken: Added the following clarification to the Methods part: "In the case of inter-dependent grouping, we also devised two subgroups. A context-rich subgroup containing the inter-links where each of the proteins in a PPI is supported by at least two inter-linked lysines involving different reactive sites on both ends between the same proteins and a second, context-poor, subgroup containing all other inter-link matches."

The legend to Figure 1a says "... proteins were denatured and cross-linked in groups of eight ...", citing Ref. 17. When checking Ref. 17, it reads "The dissolved proteins were mixed in pairs of two proteins in all possible combinations within one interaction group. They were incubated for 20 min at 50 {degree sign}C to induce interactions in vitro. 0.2- 1 mM DSSO cross-linking reagent was added to the groups and incubated at room temperature for 30 min." Please clarify.

Thanks for this comment! We agree that this experiment can be described better. The current phrasing suggests that eight proteins are being cross-linked, but actually proteins are allocated into groups of 8, which were then mixed and cross-linked in all possible pair-wise combinations.

Action taken: We rephrased our legend to reflect the wording of the original citation: "Following pairwise mixing of recombinant proteins within one interaction group, proteins were denatured, cross-linked, followed by MS analysis."

Reference #1 needs to be cleaned up.

Nice catch! Cleaned up.

Plots in Fig. 4 and Extended Data Fig. 4 use different color schemes, which is a bit confusing.

We adapted the color schemes of Extended Data Figure 4 to reflect the coloring in Figure 4.

Reviewer #2:

Summary:

This paper explores the use of different strategies to assess and control false discovery rate (FDR) in cross-linking mass spectrometry (XL-MS) studies, and proposes a new strategy for higher quality XL-MS datasets. As erroneous identifications usually present as inter-protein cross-links, this is of high interest for large scale XL-MS studies where high confidence protein-protein interaction mapping is often the goal. The authors proposed a new strategy for FDR control which combined two main steps: (1) the use of a target-decoy fusion database (as opposed to the usually used concatenated target-decoy databases), and (2) the grouping of cross-linked spectra populations by level of parallel observations as well as by cross-link information type (intra- vs inter-link) BEFORE the FDR control takes place. This enabled effective control of false positives whilst substantially decreasing the rate of false negatives, producing more high quality cross-links that represent bona-fide PPIs. The authors assessed this FDR strategy (alongside several others suggested and used by the field) using a unique ground-truth dataset (described in another paper by the same research group), a synthetic dataset, and real XL-MS datasets from different organisms. They also investigated how it's use improved interactome coverage and the interpretation of structural models of protein-protein interactions.

Overall, the paper is well written. The findings presented by the authors are exciting and present a key conceptual and technical finding that would be of very high interest and utility for the XL-MS community.

Thanks! We appreciate this comment!

Minor points:

If implemented by popular XL-MS search engines, the strategy should improve the sensitivity and accuracy of future XL-MS studies. The authors show that this strategy works on posthoc FDR analyses from cross-link data produced by XlinkX, which does not have the fusion/context-subgrouping strategy in-built. Do these findings generalise to the identifications produced from other search engines such as pLink2, MSAnnika, MeroX, etc? How much better does the FDR control get beyond what is performed by default in these other XL-MS programs?

The reviewer is right. Testing other software is relevant for estimating the robustness of our approach. pLink2 analyzes non-cleavable cross-linkers and is therefore not suited for comparison with the DSSO/DSBSO datasets used in this study.

We have analyzed the performance of the softwares mentioned by the reviewer in the ground-truth dataset used in this study and developed in Clasen and Ruwolt et al. (Nat. Methods, accepted, see pre-print here: <https://doi.org/10.1101/2023.11.30.569448>). Figure 3 of the pre-

print evaluates the recall of false inter-link IDs in MeroX, MSAnnika and MaxLynx. This shows that MSAnnika performs overall comparably robust FDR control at the CSM and residue-pair level, while MeroX fails to properly do so, even at the CSM level. For this reason, we evaluate the benefit of context sensitive strategies using MSAnnika only. We focused on the dataset and strategy that gave overall the highest benefit compared to the standard inter-intra separate search (HEK293T and inter-dependent grouping). We first calculated the CSM FDR at inter-link level and collected those inter-CSMs surviving an arbitrary 10% CSM FDR cut-off that was implemented as prior filtering (see Reviewer Figure 5a). Following aggregation into residue-pairs, we grouped residue pairs into a context-rich and context-poor subgroup based on parallel inter-link observations (inter-dependent subgrouping, see Reviewer Figure 5b). Following assembly into a combined list, we evaluated the performance across different target-decoy FDR thresholds for increasing sensitivity. This shows that sensitivity can be increased by ~75 % at a 1% FDR cut-off (Reviewer Figure 5c), which is consistent with the increases in sensitivity we observed using XlinkX in our original analysis.

Reviewer Figure 5. Evaluation of inter-dependent subgrouping to data searched by MSAnnika. (a) Prefiltering step at inter-CSM FDR of 10% (dotted line), followed by residue pair aggregation and inter-dependent subgrouping (b). (c) Comparison in recall of target inter-links across several FDR cut-offs comparing the standard inter-intra separate search and interdependent subgrouping at the level of recalled inter-links. The dotted line indicates a target-decoy based FDR cut-off at 1 %.

This demonstrates that our results hold true for data searched by other software. We include the R scripts used to analyze MSAnnika data in the github repository (see also next comment).

Although not necessary, one way to greatly improve the study's direct usefulness for the field would be to include a tool (perhaps even just an R script) that can (1) generate the fused target-decoy FASTA files, and (2) perform their subgrouped post-hoc FDR analysis on the output of commonly used search engines. Such a tool should report the error at all levels of biological redundancy (spectral match, peptide pair, residue pair, protein pair).

The reviewer is right, such a tool would indeed be very helpful for the community. We think that such a tool should be easy to use and perform FDR control at all levels, as the reviewer

mentioned. At best it is also embedded in a robust and fast computational environment. Although an R script is not the best option when it comes to these points, we make the R script available that was used for analyzing MSAnnika data (see also **Reviewer Figure 5**)

To address the reviewer's point more sustainably and enable usage of this strategy in the community, our software development team is starting to implement it into our high-speed search engine for cross-links, Scout, which was recently accepted in Nature Methods (see pre-print: <https://doi.org/10.1101/2023.11.30.569448>). We are committed to providing context sensitive strategies as an alternative option for FDR control in the next Scout update. However, such a software update requires major rearrangements in the C Sharp/Python architecture in Scout and is beyond the scope of this revision, because it requires (1) several additional months of work and (2) transferring the project to the Scout software developers who have not been involved in this manuscript before.

Action taken: The R script used for analyzing the MSAnnika data has been made available at https://github.com/Bogdanob/XLMS_decoyFusion/.

Given its importance for the paper, the explanation of the benefits of using a target-decoy fusion over a concatenated target-decoy database could be explained better. After some reading, I now understand the motivations for its use, but just a few more sentences explicitly stating why it would better model the error for the subgroup FDR strategy would greatly increase the readability of the paper. On a similar note, the conceptual explanation of how the synthetic dataset was generated could be clearer.

Thanks for the critical feedback! We added the following sentences to the Discussion part on the advantage of decoy-fusion:

“The use of decoy fusion databases upon context-sensitive subgrouping is critical as it will model the error rate more faithfully. Decoy fusion ensures that both the target and decoy complements are placed in the same subgroup, whereas in concatenated strategies, they might end up in different subgroups.”

Additionally, we added the following sentences to the Results part when introducing the synthetic dataset:

“Since the true number of false positives in the HEK293T data cannot be known, we simulated the theoretically expected distribution of true and false positives in the context-rich and context-poor subgroups based on a set of simple assumptions and controlled parameters (see Methods). We assume that wrong matches are randomly assigned to any protein in the database, while true matches have the tendency to frequently match to a restricted set of proteins. Following the placement of correct and incorrect matches on the proteins, we grouped context-poor and context-rich inter-links and evaluated the fraction of correct and incorrect matches in these subgroups. For intra-dependent grouping...”

Additionally, we updated the corresponding Figure (Figure EV2), which shows the workflow for creating/analyzing the “synthetic” dataset.

Reviewer #3:

This manuscript is a timely addition to advancing crosslinking MS analysis beyond studying individual protein complexes toward whole proteomes. The authors address the challenge of balancing sensitivity and specificity in identifying protein-protein interactions (PPIs) by developing a target-decoy fusion strategy that allows context-dependent data filtering. Before this study a few groups have suggested ways to filter on protein groups and have shown impressive results, however, as the authors discuss, filtering on the protein level using a concatenated target-decoy database can cause the link between targets and decoys to be lost. To address this the authors have used a fused-decoy strategy like the one (recently preprinted) by Fischer and Rappsilber. Together these manuscripts demonstrate the current way to subgroup crosslinking MS data on the protein level and I enthusiastically support this manuscript for publication.

We thank the reviewer for the positive comment!

Comments:

From reading the manuscript I think that the main gain from the fused-decoy approach is that crosslinks can now be filtered at the protein level. This made me consider whether the context-dependent approach suggested by the authors is the optimal use of information in the real biological datasets. I would like the authors to try to look at the effects of protein abundance information instead to see how it compares.

Typically, before crosslink search on a proteome-scale sample it is standard practice to filter the proteins searched by an abundance cutoff from standard proteomics. This dramatically helps boost matches as crosslinked peptides tend to only be found in the most abundant proteins. My concern with that approach has always been that linear (and linear-modified) peptides from proteins that aren't included in the search can be matched as decoys or falsely matched as crosslinks. I would like to see if the authors' fused-decoy approach can allow filtering on protein abundances *after* search. I would guess that the chances of an identified crosslink being 'real' increase at higher protein abundances.

My thinking is that protein intensity filtering is already happening with the context-dependent approach anyway, with proteins counted as 'context-dependent' being much more abundant than those in the other group. It would be useful if the authors checked this in the HEK dataset and any others.

The reviewer has a very good point! The proteins in context-rich subgroups are probably also quite abundant. We would also like to point out that abundance is an important but not the only

predictor for the formation of cross-links. For example, this also includes structural complexity of the protein (e.g. a disordered protein giving more cross-links than a rigidly structured protein). Additionally, sequence properties play a role such as the overall sequence length, availability of reaction sites and protease cleavage sites. While a context-sensitive approach on the basis of cross-links accounts for all these aspects, limiting the analysis to protein abundance may not account for all the other aspects.

To directly evaluate the idea of the reviewer we evaluated the mitochondria dataset (Zhu et al., Nat Commun, 2024), for which we quantified overall protein abundance level (iBAQ) using MaxQuant. We first categorized the top 5% of identified proteins into a high protein abundance group based on the iBAQ and the remaining into a low protein abundance group (**see Reviewer Figure 6a**). Based on this, we then categorized inter-links into a high XL-abundance group when both the linked proteins were in the high protein abundance group. The remainder went into the low abundance group, resulting in two inter-link subgroups with balanced XL-count (**Reviewer Figure 6b**). Importantly, this was done in a fused design, so that decoy entries where the high-abundance target counterpart was matched are rescued. We then combined both subgroups and performed FDR control, as explained in the subgrouping and FDR section in our manuscript. We observed an increase in sensitivity across all relevant FDR thresholds (0-5%). At a stringent FDR cut-off at 1 %, we observed an increase in sensitivity of 27 % (**Reviewer Figure 6c**), which is only minimally worse than the improvements we observed with inter-dependent subgrouping for this dataset (29 %).

While these initial results are certainly interesting and promising, we feel that substantially more work needs to be done to thoroughly evaluate the potential of this workflow, including, the definition of appropriate cut-offs for low and high-abundance groups. Further, this workflow requires external information (protein abundance levels) to be considered, which makes it more laborious than relying on cross-link information only.

We therefore feel that including these data is beyond the scope of our manuscript.

Action taken: To acknowledge that protein abundance may be an important aspect that is worth considering, we added the following sentence in the Discussion section:

“Fused target-decoy strategies may also be used to consider other non-XL based protein information, such as protein abundance to increase sensitivity.”

Reviewer Figure 6. Considering protein abundance for context-sensitive subgrouping improves sensitivity. (a) proteins are grouped into a high and low abundance group based on MaxQuant reported iBAQ values. (b) inter-links are grouped into a high abundance group when both inter-links match to a protein within the high abundance group from a. (c) Recall of target inter-links as a function of the FDR cut-off comparing a subgrouping strategy using the abundance grouped inter-links (orange) compared to a standard inter-intra separate strategy (grey). The dotted line indicates a target-decoy based FDR cut-off at 1 %.

The size of each subgroup is important information if there is to be an accurate estimation of FDR based on the decoys remaining. The percentages and ratios in the figures are important, but it is also useful for the reader to know the number of CSMs, residue pairs, and proteins in each subgroup.

We reported the sizes of each of the subgroups in our original manuscript at the level of residue-residue pairs.

Action Taken: We added the number of CSMs and proteins within all the subgroups (new Figure EV3h-j and Figure EV4).

The methods are missing two key pieces of information that would allow interpretation of the results, 1- that the data was acquired as MS2, and 2- how XlinkX handles merging from CSMs up through the levels to crosslinked residue pairs. I assume only top-scored CSMs are used for each spectrum throughout. Also, I assume when crosslinks are discussed in the manuscript that refers to 'unique residue pairs'?

The reviewer is correct - the top scored CSM is reported as the representative for the ResPair.

Action Taken: We added the information that data were acquired as MS2, when describing the HEK293T and ground-truth dataset in the methods section. Further, we

clarify that “cross-links” refers to unique residue pairs and that the top score CSM is reported as the representative for the ResPair.

I do not understand the step of merging of the subgroups to generate a final combined list. If each subgroup had 1% FDR and they are truly separate groups (no overlap of matched spectra) why can't they just be merged and the FDR percentage maintained. Combining the results post 5% FDR should still give you 5% FDR in the combined list. If you get different results after combining something is off.

The reviewer has a good point that we also thought about for some time but reached a different conclusion than the reviewer. First, it is important to emphasize that FDR controls the fraction of false positives in the group you want to have consolidated. In our case, we chose to consolidate all inter-links. As correctly pointed out by the reviewer, a FDR cut-off of 5 % at inter-link level should limit the fraction of decoys among all inter-links to 5 %. To ensure that this is always the case we perform this harmonization step that entails merging the subgroups into a final list on which the overall FDR is calculated. Not doing this harmonization step would result in problems in certain situations:

Consider the following hypothetical example: An inter-link dataset of 4000 targets and 600 decoys. This gives an overall FDR of 15% in the total list. Upon context-sensitive subgrouping this list is split into two groups. One context-rich with lower decoy count (e.g. 100/2000, 5% FDR) and one context-poor with higher decoy count (500/2000, 25 % FDR). In this example, the reviewer's strategy is identical to ours in the range where FDR can be reported in both subgroups (up to 5 %). However, this is different when e.g. a 10 % cut-off is desired. Then all the context-rich are accepted but only a fraction of the context-poor. This results in overall fewer decoys among inter-links reported then would be required to reach 10 % FDR. The situation is even more pronounced when a cut-off of 20 % is imposed.

*To substantiate this with a real-world example, we show the analysis from the HEK293T dataset comparing the standard inter-intra separate to the inter-dependent subgrouping strategy with or without harmonization (**Reviewer Figure 7, left**). In this case, both unharmonized and harmonized strategies give similar FDR estimates in the range where both are reported (0 - 0.6 %, **Reviewer Figure 7, right**). However, at higher FDR cut-offs both strategies disagree, with the harmonization strategy giving the accurate account of the total error among all inter-links.*

Reviewer Figure 7: Harmonization of subgroup FDRs assures accurate reporting of FDR among all inter-links in the HEK293T dataset. (left) Number of recalled inter-links as a function of the applied FDR cut-off. Note that the standard and subgroup-harmonization strategy both end in the same number of recalled inter-links at identical FDRs, reflecting the overall decoy count among all inter-links. This estimate is skewed when no harmonization is performed. Due to relatively higher FDRs in context-poor subgroups, higher FDR cut-offs are required to recall the same number of inter-links. Note that the FDR-values for the not harmonized strategy are not accurate and do not reflect the actual fraction of decoys among all inter-links. **(right)** Direct comparison of the imposed FDR cut-off (x-axis) to the calculated FDR in the total list of inter-links surviving the FDR cut-off, separately for the harmonized or non-harmonized strategy. FDR cut-offs imposed on the unharmonized subgroups do not always reflect the overall error among the total set of inter-links.

We would also like to point out that we were not the first to perform harmonization strategies following subgrouping. For example, our approach is similar to how shotgun proteomics softwares such as MaxQuant handle diverging error rates between peptides of different length. First, so-called PEP (posterior error probability, see Figure 4 of the Cox and Mann, 2008, see <https://doi.org/10.1038/nbt.1511>) values are calculated based on the decoy and target distributions from peptides with the same length. Those PEP values from various length-dependent subgroups are then integrated into a final list, before the actual FDR is calculated based on the PEP values.

However, we agree with the reviewer that our description and workflow may be confusing. This is mainly because FDR is a population-specific statistic, while the subgroup FDR values we used are identification-specific. In order to avoid this confusion for the readers of the manuscript we now implemented posterior error probabilities (PEPs). Following subgrouping and PEP calculation in the individual subgroups, the PEPs are combined into a combined list on which the FDR calculations are performed.

We thank the reviewer for pointing out this potential source of confusion.

Action Taken: Implemented PEP values to facilitate merging of identifications from different subgroups, which is described in the Subgrouping and FDR calculations part within the Methods.

Another issue is that if one uses false discovery rates of each match in the merging you only have the resolution of the numbers of decoys you have remaining. If the overall numbers are quite small you get large jumps in FDR between matches.

We are not sure whether we completely understand the reviewer. It appears that the reviewer is concerned that the merging (see previous comment) results in few remaining decoys and that this then limits the possibility to calculate accurate FDRs.

First, the merging of the subgroups does not result in fewer decoys among inter-links. The overall number of decoys is preserved with our strategy and only their relative position within the sorted list on which FDR is performed is different to the standard, intra-inter separate search.

Nevertheless, it is still possible that large jumps of accepted inter-links at specific FDR values occur, which may limit the resolution of our FDR calculations in specific ranges. To mitigate this issue we have now implemented PEPs (see previous comment to the same reviewer) that are based on modeling target-decoy distribution within the subgroups and are then combined and used for overall FDR control. Please also note that this did not affect any conclusions or trends that we reported.

Action Taken: We implemented PEP values to facilitate merging of ResPair lists from different subgroups.

Minor comments

- Please use HEK293T instead of HEK throughout.

Done.

- The figures could be improved with better labeling. Particularly the numbers on and inside bars.

We increased the font size on labeled bars in Figure 3 and Figure EV3.

- With the controlled dataset the authors have, have they considered other ways of filtering such as those discussed in the introduction, such as different sizes of crosslinked peptides?

This is a good idea! We have indeed observed in various datasets that peptides with shorter lengths more frequently matched to decoys than peptides with longer lengths, but we did not analyze this effect systematically. This is consistent with previous observations made by others

in shotgun proteomics (see e.g. Cox and Mann, 2008, Nat Biotech, <https://doi.org/10.1038/nbt.1511>). This opens the possibility to make use of these differences for adjusting the posterior error probabilities of peptides with different lengths for XL-search engines. Agreeing with the reviewer, we are in the process of incorporating peptide length as a feature in the next version of our newly developed Artificial Neural Networks -based search engine Scout (just accepted in Nature Methods, preprint can be read here, <https://www.biorxiv.org/content/10.1101/2023.11.30.569448v1>). We hope the reviewer will understand that finalizing this update of Scout is beyond the scope of this revision (see also our response to Reviewer#2's second minor comment).

- The HEK dataset give 12,216 targets and only 2003 decoys without FDR filtering. This seems low, but I assume only spectra with DSSO doublets are reported as matches?

Yes, this is correct. Also, there is a pre-filtering step before reporting CSMs in XlinkX, that is, at least three out of four DSSO-cleaved signature peaks are observed in the corresponding MS2 spectrum.

8th Nov 2024

Manuscript Number: MSB-2024-12447R

Title: Redesigning error control in XL-MS enables more robust&sensitive protein-protein interaction studies

Author: Boris Bogdanow

Max Ruwolt

Julia Ruta

Lars Muehlberg

Cong Wang

Wen-Feng Zeng

Arne Elofsson

Fan Liu

Dear Fan,

Thank you for sending us your revised manuscript. We have now heard back from the three reviewers who were asked to evaluate your revised study. As you will see below, the reviewers are overall satisfied with the performed revisions and support publication. Before we can formally accept the manuscript for publication, we would ask you to address some remaining issues listed below:

1. Regarding the citation of related work from the Rappsilber group, we suggest removing the statement about the timeline of both preprints and including the following sentence only: 'This is consistent with a similar strategy by the Rappsilber lab to restore error control of the mi-filter (Fischer & Rappsilber, 2024).'
2. Please remove the "Author contribution" section from the manuscript file.
3. Please provide up to five keywords in the manuscript file.
4. Funding: please ensure the following funding information is included in the manuscript file: Deutsche Forschungsgemeinschaft (DFG) LI 3260/6-1.
5. The title of conflicts of interest statement should be renamed to "DISCLOSURE AND COMPETING INTERESTS STATEMENT".
6. Please upload Appendix file as a PDF file.
7. Data availability section
 - Please include the "Code availability" in the Data Availability section.
 - Please note the data availability is restricted to new primary data that are part of this study. Data not produced in the current study should be removed. Additionally, update the 'data availability' information in the author checklist accordingly.
 - You are encouraged to include *data citations in the reference list* to directly cite the datasets that were re-used and obtained from public databases. Data citations in the article text are distinct from normal bibliographical citations and should directly link to the database records from which the data can be accessed. In the main text, data citations are formatted as follows: "Data ref: Smith et al, 2001". In the Reference list, data citations must be labeled with "[DATASET]". A data reference must provide the database name, accession number/identifiers and a resolvable link to the landing page from which the data can be accessed at the end of the reference. Further instructions are available at .
8. Please provide a Reagents and Tools Table as a .docx file, listing key software and relevant equipment and including their sources and relevant identifiers. The aim is to facilitate adoption of the methodologies across labs. A downloadable template (.docx) for the Reagents and Tools Table can be found in our author guidelines: <https://www.embopress.org/page/journal/17444292/authorguide#structuredmethods>.
9. I have slightly modified the synopsis text(see attached); please let me know if it is fine as is or if you would like to introduce further modifications.
10. Figure Legends:
 - Please note that the box plots need to be defined in terms of minima, maxima, centre, bounds of box and whiskers, and percentile in the legends of figures 5b,e.

- Please note that information related to n is missing in the legends of figures 5b, e.

11 . Please use the following section order: title page with complete author information, abstract, keywords, introduction, results, discussion, methods, data availability section, acknowledgements, disclosure and competing interests statement, references, main figure legends, tables, expanded figure legends.

When you resubmit your manuscript, please download our CHECKLIST (<https://bit.ly/EMBOPressAuthorChecklist>) and include the completed form in your submission. *Please note* that the Author Checklist will be published alongside the paper as part of the transparent process (<https://www.embopress.org/page/journal/17444292/authorguide#transparentprocess>)

Click on the link below to submit your revised paper.

Kind regards,
Jingyi

Jingyi Hou, PhD
Scientific Editor
Molecular Systems Biology

If you do choose to resubmit, please click on the link below to submit the revision online before 8th Dec 2024.

IMPORTANT: When you send your revision, we will require the following items:

1. the manuscript text in LaTeX, RTF or MS Word format
2. a letter with a detailed description of the changes made in response to the referees. Please specify clearly the exact places in the text (pages and paragraphs) where each change has been made in response to each specific comment given
3. three to four 'bullet points' highlighting the main findings of your study
4. a short 'blurb' text summarizing in two sentences the study (max. 250 characters)
5. a 'thumbnail image' (550px width and max 400px height, Illustrator, PowerPoint or jpeg format), which can be used as 'visual title' for the synopsis section of your paper.
6. Please include an author contributions statement after the Acknowledgements section (see <https://www.embopress.org/page/journal/17444292/authorguide#manuscriptpreparation>)
7. Please complete the CHECKLIST available at (<https://bit.ly/EMBOPressAuthorChecklist>). Please note that the Author Checklist will be published alongside the paper as part of the transparent process (<https://www.embopress.org/page/journal/17444292/authorguide#transparentprocess>).
8. When assembling figures, please refer to our figure preparation guideline in order to ensure proper formatting and readability in print as well as on screen:
<https://bit.ly/EMBOPressFigurePreparationGuideline>
See also figure legend guidelines: <https://www.embopress.org/page/journal/17444292/authorguide#figureformat>
9. Please note that corresponding authors are required to supply an ORCID ID for their name upon submission of a revised manuscript (EMBO Press signed a joint statement to encourage ORCID adoption). (<https://www.embopress.org/page/journal/17444292/authorguide#editorialprocess>)
Currently, our records indicate that the ORCID for your account is 0000-0002-2358-549X.

Link Not Available

10. Include a Reagents and Tools Table as part of the Methods section, which can be downloaded from our author guidelines (<https://www.embopress.org/page/journal/17444292/authorguide#structuredmethods>)

*** PLEASE NOTE *** As part of the EMBO Press transparent editorial process initiative (see our Editorial at <https://dx.doi.org/10.1038/msb.2010.72> , Molecular Systems Biology will publish online a Review Process File to accompany accepted manuscripts. When preparing your letter of response, please be aware that in the event of acceptance, your cover letter/point-by-point document will be included as part of this File, which will be available to the scientific community. More information about this initiative is available in our Instructions to Authors. If you have any questions about this initiative, please contact the editorial office (msb@embo.org).

Reviewer #1:

I would like to thank the authors for their detailed and clear responses in the rebuttal letter.

Based on the information provided in the response to the reviewers, I conclude that all my comments have been addressed appropriately.

I agree somewhat with the authors that not all of the information provided in the rebuttal needs to go into the main manuscript because the review reports will be made public as well, but leave this decision to the editor.

Reviewer #2:

The authors have addressed all my comments to the manuscript satisfactorily. I am also satisfied upon reading their responses to the other reviewer's comments, and feel that the manuscript in this form would be ready for publishing.

Reviewer #3:

The authors have answered all the reviewer comments to my satisfaction.

In response to reviewer 1 the authors note that their manuscript was pre-printed prior to Fischer et al (now published in MSB). The authors clearly feel strongly that they should not have to cite that paper when they pre-printed first. I draw this to the attention of the editor to adjudicate what is appropriate here.

Point-by-point response to the reviewer's comments

We thank all three reviewers for the carefully evaluating our manuscript and providing feedback!

Reviewer #1:

I would like to thank the authors for their detailed and clear responses in the rebuttal letter.

Based on the information provided in the response to the reviewers, I conclude that all my comments have been addressed appropriately.

I agree somewhat with the authors that not all of the information provided in the rebuttal needs to go into the main manuscript because the review reports will be made public as well, but leave this decision to the editor.

Thanks for the overall positive comment.

Reviewer #2:

The authors have addressed all my comments to the manuscript satisfactorily.

I am also satisfied upon reading their responses to the other reviewer's comments, and feel that the manuscript in this form would be ready for publishing.

Thanks for evaluating our manuscript.

Reviewer #3:

The authors have answered all the reviewer comments to my satisfaction.

In response to reviewer 1 the authors note that their manuscript was pre-printed prior to Fischer et al (now published in MSB). The authors clearly feel strongly that they should not have to cite that paper when they pre-printed first. I draw this to the attention of the editor to adjudicate what is appropriate here.

We thank the reviewer for this comment. Following editorial suggestions, we decided to cite the reference Fischer and Rappsilber in our manuscript without commenting on the timeline of the pre-prints. Now referenced in the discussion as:

"This is consistent with a similar strategy by the Rappsilber lab to restore error control of the mi-filter (Fischer & Rappsilber, 2024)"

21st Nov 2024

Manuscript number: MSB-2024-12447RR

Title: Redesigning error control in XL-MS enables more robust&sensitive protein-protein interaction studies

Dear Fan,

Thank you again for sending us your revised manuscript. We are now satisfied with the modifications made and I am pleased to inform you that your paper has been accepted for publication.

Kind regards,
Jingyi

Jingyi Hou, PhD
Scientific Editor
Molecular Systems Biology
